

# Real-time flood forecasting with Machine Learning using scarce rainfall-runoff data

Théo Defontaine[1], Sophie Ricci[1], Corentin J. Lapeyre[1], Arthur Marchandise[2], and Etienne Le Pape[3]

[1]CERFACS-CECI, UMR 5318, 31057 Cedex 1, Toulouse, France
[2]Direction des Risques Naturels, DREAL Occitanie, 31074 Cedex 9, Toulouse, France
[3]Service Central d'Hydrométéorologie et d'Appui à la Prévision des Inondations, Ministère de la transition écologique, 31057 Cedex 1, Toulouse, France

**Correspondence:** T. Defontaine (defontaine@cerfacs.fr), S. Ricci (ricci@cerfacs.fr), C. J. Lapeyre (lapeyre@cerfacs.fr)

**Abstract.** Flooding is the most devastating natural hazard that our society must adapt to worldwide, especially as the severity and the occurrence of flood events intensify with climate change. Several initiatives have joined efforts in monitoring and modelling river hydrodynamics, in order to provide Decision Support System services with accurate flood prediction at extended forecast lead times. This work presents how fully data-driven Machine Learning models predict discharge with better

performance and extended lead-time, with respect to the current empirical Lag and Route model used operationally at the local Flood Forecasting Service for the Garonne River in Toulouse. The database (DB) is composed of discharge and rainfall data, upstream of Toulouse, for 36 flood events over the past 15 years (40k data points). This scarce data set is used to train a Linear Regression, a Gradient Boosting Regressor and a Multilayer Perceptron in order to forecast the discharge in Toulouse at 6-hour and 8-hour lead times. We showed that the Machine Learning approach outperforms the empirical Lag and Route for 6-hour

lead-time. It also provides a reliable solution for extended lead times and saves the implementation of a new empirical Lag and Route model. It was demonstrated that the scarcity and the heterogeneity of the data heavily weigh on the learning strategy and that the layout of the learning and validation sets should be adapted to the presence of outliers. It was also shown that the addition of rainfall data increases the predictive performance of Machine Learning models, especially for longer lead times. Different strategies for rainfall data pre-processing were investigated. This study concludes that, with the present test case,

time-averaged rain information should be favored over instantaneous or time varying data.

## 1 Introduction

Flooding is one of the most common major natural disasters[1]. It can occur at local to medium scale, and is closely linked to the water cycle and its variability. As such, climate-change induced perturbations can potentially aggravate this risk that threatens populations and infrastructures. Emergency services in charge of surveillance and alert (https://emergency.copernicus.eu) rely

on forecasting and risk assessment to effectively mitigate flood damage and their impact on financial markets (Bressan et al., 2022; Jongman et al., 2014; Campbell Johnston et al., 2015). As stated by the United Nations Office for Disaster Risk Reduction in the recent Global Assessment Report (for Disaster Risk Reduction, 2022), 'The best defence against future shocks is to

---

[1]https://www.prevention web.net



transform systems now, to build resilience by addressing climate change and to reduce the vulnerability, exposure and inequality that drive disasters.'

Emergency services monitor and forecast floods to better protect human and economical assets (Wu et al., 2020) based on algorithmic and statistical solutions that quantify flood risk (Woodward et al., 2011). Decision support systems allow to identify high-risk areas and elaborate emergency procedures to improve awareness and preparedness of governmental agencies and disaster managers.

Assessing the potential damage of a flood event in the past or in the future is a major component of flood risk management
(Kron, 2002). It allows to allocate resources for improving monitoring and prediction of the flood hazard. It also enables informed development of infrastructures for protection and resilience. To quantify the economic impacts of flooding, one may note the ECLAC method in Mexico (CEPAL, 2007) and the Hazus method (FEMA, 2018) used in the US and in Canada. It specifically uses a flood-depth model and a damage module, in the purpose of locally estimating loss. This is for the benefits of federal, state, regional and local governments, and private companies that plan risk mitigation, emergency preparedness,
response and recovery. A review (Adeel et al., 2020) of comprehensive methods for evaluating the economic impacts of floods in Canada, Mexico and the United States presents conclusions from a collaborative research project initiated by the Commission for Environmental Cooperation that brought together government agencies, academic institutions and stakeholders from the private sector. This review focuses on the assessment of direct damages to property and infrastructures as well as indirect losses triggered by disruption of routine activities. Model-based tools such as HEC-IFA (USACE, 2015) and HEC-FDA (USACE,
2016) from the United States Army Corps of Engineers, combine hydrodynamic models that represent the physical hazard with discharge or water level estimates, with economical models that represent associated loss at local or large scale. Model-based methods take into account multiple variables such as elevation, topography, economic losses, building characteristics, etc., to estimate flood depth, duration, location, and damage to various sectors. Determining direct flood damage is commonly done using depth-damage curves, which denote the flood damage that would occur at specific water depths per asset or per
land-use class (Huizinga et al., 2017). At larger scale, the Computable General Equilibrium models use a series of equations to summarize the market dynamics, calibrated by empirical economical data to estimate how an economy might respond to changes in policy and technology.

Decision support services make the most of Earth observations from different types of sensors (Kruczkiewicz et al., 2021) such as in-situ observing stations, land-based radar observations, airborne measurements or remote sensing satellite data from
Earth Observation programs (i.e. SAR, optical, nadir or large swat altimetry instrument on board). This constitutes a rich and ever increasing volume of heterogeneous data for continental water, with various precision, resolution, frequency and spatial coverage. While in-situ stations are easy to install, they are prone to impairments, costly to maintain, and only provide a poor spatial coverage. Airborne observation is well adapted to event-based monitoring, measurement campaign, punctual in time and space by definition and thus not commonly used for operational forecast or surveillance. Radars provide real-
time and spatially dense measurements at global scale to complement land-based observations with high precision. They provide accurate information provided calibration by land-based data, as presented by Laurantin (2013)'s rain product. Yet they may be hindered by elevated terrains and land cover. Earth observation from space provide frequent global images of





water data of various kinds (rainfall, extension of water bodies, soil moisture, topography, water surface elevation, vegetation
,...). This gigantic volume of data from space, while it remains imperfect and comes at high cost, offers an unseen opportunity
to acknowledge the dynamics of continental waters. Nevertheless, the capacity to forecast flood with space data for small
to medium catchments remains limited by the revisit period. This limitation is mitigated when considering multi-mission
products (Schumann, 2021). The scarcity and heterogeneity of the observations are also managed by the use of numerical
models, eventually with data assimilation, that come as an interpolation tools over space and time for past periods, and as
extrapolation tools for futur periods. They are denoted as Flood Forecasting Models (FFMs) in the following.

The accuracy of numerical models greatly impacts the capability of emergency services. Their use strongly relies on the
availability of topographic, hydrological and hydraulic data as well as computational resources. The dynamics of continental
water can be solved at different temporal and spatial scales, taking into account the associated physical processes and data.
Typical hydrological rainfall-runoff models chain a land surface model that simulates the energy and water balance at the
soil-atmosphere-vegetation interface with a river routing model that emulates the lateral transfer of freshwater towards the
continent-ocean interface, usually solving spatial scales ranging from 5 km to 50 km with temporal scale from a few days
to a few months. In order to take into account smaller scales and complex processes, hydraulic models solve the Shallow
Water equations (down to meters and minutes), at the expenses of a significant computational cost and provided a precise
knowledge of topographic, bathymetric and vegetation data. Both hydrological and hydraulic models require a calibration step,
using observational data sets for reference events. In spite of calibration, the remaining uncertainty in hydrologic and hydraulic
model inputs and numerical schemes translates into uncertainty in the simulated water level and discharge; these uncertainties
can be reduced with data assimilation algorithms for real-time forecast.

Fully data-driven hydrology models come as an alternative, for instance when topographic/bathymetric terrain data are not
available. While these models can be simple and adaptive, they usually struggle to make out-of-the-box predictions (for events
that do not appear in the DB) and require a large and various learning data set. A large variety of FFMs based on Machine
Learning (ML) were reported recently in the literature. These approaches were implemented for studies over a great number
of catchments worldwide, within operational framework in some cases (Yuan et al., 2020; Nevo et al., 2022). Most strategies
are based on Neural Network (NN) models such as Multilayer Perceptron (MLP) (Riad et al., 2004; Mosavi et al., 2018;
Noymanee and Theeramunkong, 2019), which is a simple version of feed-forward neural networks  (Dawson and Wilby, 1998;
Toukourou et al., 2011); or ANFIS models (Khac-Tien Nguyen and Hock-Chye Chua, 2012; Nguyen et al., 2014). These studies
showed that ML models can outperform standard solvers like rainfall-runoff models, especially for highly correlated networks
and short lead times. Gradient Boosting Regressors (GBRs) are known to be well adapted to temporal data. Venkatesan and
Mahindrakar (2019) showed that a better accuracy was obtained with a GBR (XGBoost) than with a random forest or a
Support Vector Machine (SVM) for short-term flood forecasts. Sanders et al. (2022) presented the implementation of GBRs
for short term operational flood forecasts. Recent publications also showed the use of advanced NN models such as Recurrent
Neural Networks (RNNs) and more specifically Long Short-Term Memory networks (LSTMs) (Song et al., 2020). RNNs
are well adapted for flood forecasting as they are devised to deal with time-series. The LSTM-based FFM was developed
by Kratzert et al. (2018) and applied for large-scale studies (Nearing et al., 2021), over US catchments (Kratzert et al., 2019a, b),





outperforming both local and regional models. This work was recently extended to 48 countries worldwide as shown in Kratzert et al. (2022). It should be noted that LSTMs require large datasets and extensive computational resources, as argued by Kratzert
et al. (2023).

When large data sets are not available, data-driven strategies should be adapted to the scarse-data layout. Basic empirical models based on lagged hydrographs (Vidal et al., 1998; Tiantian et al., 2020), or classical Linear Regression (LR), usually rely on in-situ measurements. They are typically calibrated over small catchments (Jun et al., 2016). When the data set presents as time-series, the temporal coherence of the signal may be accounted for, for instance with wavelet analysis (Adamowski, 2008).
Box-Jenkins or auto-regressive integrated moving average (ARIMA) models (Toth et al., 1999; Wei Ming, 2020) are based on the decomposition of time-series in linear (or quasi-linear) bases.

The present study focuses on real-time flood forecasting with ML using scarce and heterogeneous rainfall and runoff data over the Garonne catchment, upstream of Toulouse. Over this catchment in the Pyrenees foothills, the geometry of the river bed and the floodplain strongly varies, thus making the implementation of a hydrodynamic model difficult. The local Flood
Forecasting Service (FFS) [2] in Toulouse has implemented an empirical forecasting model based on a linear combinaison of lagged hydrographs measured at upstream hydrometric stations (named empirical Lag and Route (ELR)). This upstream part of the Garonne River is landlocked by surrounding mountains, and the flow is quasi-linear, so that the linear assumption remains valid for short lead times. The ELR model was calibrated from gauge measurements during minor to major flood events for 4-hour and 6-hour lead times. The scarcity of the data, forged with a limited number of extreme events, only allows
for the calibration of a simple model with limited number of parameters. The accuracy of such models remains limited when important events occur, for instance when local rain represents a significant contribution to the flood. Most importantly, the extension of the forecast lead time beyong 6-hour would require the implementation and calibration of a new ELR model including additional upstream stations. As an alternative to this ELR approach, the use of models based on ML is investigated here. Among the commonly used methods in the literature, a LR, a GBR and a MLP are used due to their compatibility with
small databases. A preliminary study (Defontaine et al., 2023) showed that some ML models outperform ELR when a limited number of discharge data from significant flood events is used for 6-hour forecasts. In the present work, the DB is completed with additional flood events and rainfall time-series. Indeed, it appears that the impact of rainfall on forecast discharge increases with the lead time and that rain should be taken into account as an input data. Yet, the scarcity and the heterogeneity of the data still heavily weigh on the learning strategy.

The remainder of this article is organized as follows. Sect. 2 describes the Garonne catchment near Toulouse and the data available for the learning task. The ML models and their implementation are described in sect. 3, along with the validation criteria used for assessment. Predicted discharges are presented in sect. 4 for 6-hour and 8-hour forecast lead times. Sect. 5 finally gives some conclusions and perspectives for this work.

---

[2]SPC, Service de Prévision des Crues





## 2 Study material

### 2.1 Operational discharge forecast at Toulouse's Flood Forecasting Service

The study area is a $10133.95$ km$^2$ watershed of the Garonne river and its tributaries, shown in Fig. 1. It extends from the Pyrenees to the hydrometric station Toulouse Pont Neuf (TPN). This catchment is covered by several observing stations of the Vigicrue network[3] that provides water level observations (noted $h$, in meters) every 15 minutes. These measurements are converted into discharge (noted $Q$ in m$^3$.s$^{-1}$) with a local rating curve built from a limited number of $(h,Q)$ gauge measurements, and usually extrapolated for high flow. This part of the valley is identified as an area at high risk of flooding, as significant floods have affected it in 1875, 1930, 1952, 1981, 2003, and more recently in January 2022. This catchment was equipped in the nineteenth century with infrastructures to protect the Garonne plain downstream of Toulouse from flooding. The position of the Vigicrue in-situ stations was mostly decided after significant flooding events originating from the Gers catchment (North). As of today, experts at local FFS consider that this network is too scarce, especially to monitor events that may originate from other weather influences, which tends to occur more frequently with climate change (Caballero et al., 2007; Grusson et al., 2018; Dumas et al., 2013).

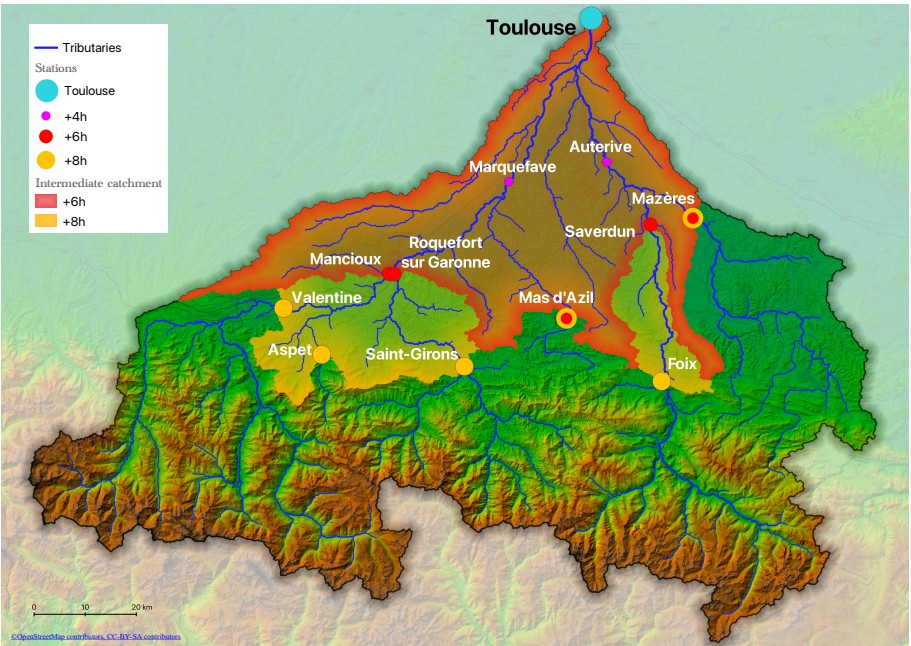

**Figure 1.** Garonne Catchment and its tributaries (in blue). Colored dots indicate the location of Vigicrue hydrometric stations used for ELR at $\tau = +4$ h forecast (purple), $\tau = +6$ h forecast (red) and $\tau = +8$ h forecast (yellow).

---

[3]www.vigicrue.fr



The Garonne River and its tributaries upstream of Toulouse are represented in blue in Fig. 1. The Vigicrue observing stations are indicated with color-coded dots with respect to the approximate propagation time to Toulouse. Toulouse's FFS devised and calibrated an empirical Lag and Route (ELR) model for +4 h and +6 h forecast, working from discharge anomalies between upstream observing stations and TPN. In the following, time is indicated by $t$ and lead time is noted $\tau$. The observed discharge is noted $Q_{loc}$ where $loc$ is an observing station of a list {Net}[4] from the observing network. The discharge anomaly at $loc$ between current $t$ and $\tau$ is noted $\Delta Q_{loc}(t) = Q_{loc}(t+\tau) - Q_{loc}(t)$ where $t$ varies over observed measured times for significant events with $\tau = +4, +6, +8$ in the present study. The predicted discharge in TPN is noted $\widetilde{Q}_{tpn}$. The ELR predicts the discharge anomaly in TPN noted $\Delta Q_{tpn}$, it is based on a calibration using the tributaries hydraulic reach over measurements made at $N_{loc}$ stations of the observing network. It reads:

$$\Delta \widetilde{Q}_{tpn}(t) = \sum_{loc\ \in\ \{Net\}} a_{loc} \Delta Q_{loc}(t - \tau). \tag{2.1}$$

The $\tau = 4$ h (noted $\tau 4$) ELR model is based on discharge measurements at Marquefave and Auterive. These constitute the {NQ4}={Mq; Au} list of observing stations, represented with purple dots in Fig. 1. It is here assumed that the effect of rainfall between these stations and Toulouse is negligible. The $\tau = 6$ h (noted $\tau 6$) ELR model from FFS is based on measurements at Mancioux, Roquefort-sur-Garonne, Mas d'Azil, Saverdun and Mazères that compose {NQ6}={Mn; RG; MA; Sv; Mz} (represented with red dots in Fig 1). As the non-linearity of the flow increases with the distance between the upstream stations and TPN, the $\tau 6$ ELR model is more difficult to calibrate and underperforms with respect to the $\tau 4$ ELR model. Additionally, the assumption of negligible local rainfall may no longer hold for strong events. This advocates for an advanced modeling strategy that takes into account the rain over the intermediate catchment upstream of Toulouse represented in red in Fig. 1. This also advocates for an alternative modeling strategy for extended lead time forecasts. Defontaine et al. (2023) considered the FFS's $\tau 6$ ELR model as baseline and showed the merits of ML approach, learning from {NQ6}. FFS ELR models for $\tau 4$ and $\tau 6$ were calibrated with 20 high flow events between 2005 and 2015 with various characteristics of severity, origin and dynamics. Both $\tau 4$ and $\tau 6$ ELRs are used by Toulouse's FFS for operational discharge forecast at TPN. For $\tau = 8$ h (noted $\tau 8$) forecast lead times, additional upstream stations should be taken into account (Valentine, Aspet, Saint-Girons, Mas d'Azil, Foix and Mazères), building {NQ8}={Vl; As; SG; MA; Fx; Mz} (represented with yellow dots in Fig. 1). Rainfall on the intermediate catchment, delimited by a yellow interface in Fig. 1, should also be taken into account. Preliminary work at FFS devised the $\tau 8$ ELR as an aggregate of several local ELRs and the $\tau 4$ ELR. This model is not considered in the present study. We aim to demonstrate that the learning strategy may replace the implementation and calibration of new ELR model for each lead time.

## 2.2 Description of the database and data engineering

In the following, a DB of 36 high flow events (selected with a hydrometric threshold $max(h_{tpn}) > 1.70$ m at TPN over 2007-2022 is used by the ML models. Each event lasts approximately 100 hours. It should be noted that data from older events were

---

[4]specified in the following



not available from digital archives at FFS. This DB is arranged with different settings described in Table 1 and used to train, validate and test the ML models.

The ANTILOPE rainfall product (Laurantin, 2008, 2013), distributed in time (hourly before 2007, and every quarter hour after 2007) and space (squared kilometer precision), gathers radar observations and in-situ measurements. LAMEDO (Organde et al., 2013; Lamblin and Gouin, 2022) is a cumulative (hourly) and weighted product based on ANTILOPE, used at FFS. It presents time series of hyetometric information. It is used here to provide rainfall data, noted $r$, over the intermediate catchment. Hydrometric time-series from the Vigicrue network are extracted from Hydroportail[5]. The water height measurements are then

converted into discharge using local rating curves. Defontaine et al. (2023) showed that working with discharge rather than with water height measurement gives better results with ML models on this case study. In-situ stations provide $15\,\mathrm{min}$ measurements that are down-sampled hourly. For $\tau \in \{\tau 6, \tau 8\}$, the total number of available time-steps in the DB is $N_\tau = \{4404, 4002\}$ for $\tau 6$ and $\tau 8$ forecast models. For $tot \in \{F6, F8\}$, the total size of the DB is $N_{tot} = N_\tau * Z$ (20k-60k points) where $Z$ is the number of input channels.

The list of input channels includes the discharge anomaly $Q[t] - Q[t-\tau]$ for all upstream stations in {NQ6} (resp. {NQ8}) and the rain data $r[t-\nu:t]$ over the intermediate catchment. The rain is either considered as a time-series (hourly) over a period in the past $\nu \in [\![2, 24]\!]$ or as an average over this period $[t-\nu:t]$, $\nu \in [\![2, 24]\!]$, noted $\mathrm{avg}(r[t-\nu:t])$, or finally as the sole instantaneous rain $r[t]$. We note $F_{rain}$ the function that processes the rain data $r[t]$ into an input in DB:

$$F_{rain}(r(t); \nu) = \begin{cases} r[t-\nu:t] & for\ \nu \in [\![2, 24]\!] \\ r[t] & for\ \nu = 0 \\ \mathrm{avg}(r[t+\nu:t]) & for\ \nu \in [\![-2, -24]\!] \end{cases} \quad, \tag{2.2}$$

where $\nu$ is made negative to differentiate averaged from time frame cases. As the models target real-time predictions, no future rains are taken into account. The prediction of the discharge anomaly $Q[t+\tau] - Q[t]$ resulting from the FFS baseline ELR model and the LR model may also be included in the DB as an additional input data.

   The first column in Table 1 describes the type of inputs (discharge, rain and baselines for the residual models) and outputs. The second column describes at which time these data are considered. The third column gives the name of the experiment (top

row), the value of the data and the dimension of the data. The input layout is associated to the experiment id composed of an information on the lead time, noted F6 for $\tau 6$ (resp. F8 for $\tau 8$), and the process of the rainfall data, noted $r0$ for instantaneous rain, $ar\nu$ for averaged rain or $r\nu$ for time-series of rainfall data. For all experiments, the model output is the discharge anomaly predicted in TPN $\Delta Q_{tpn}(t)$. It should be noted that the choice was made not to standardize nor scale the data when not necessary, meaning only the MLP uses standardized data. The hyetometric data is multiplied by the area of the intermediate

catchment to express an equivalent discharge and homogenize the DB.

## 2.3   Database settings in a scarce data layout

---

[5]https://www.hydro.eaufrance.fr/



**Table 1.** DB inputs and outputs for different experiment settings. The first column describes the type of inputs and outputs. The second column describes the temporal layout for inputs and outputs. The third column gives the experiment id and the number of input and output channels.

| | time | t-8h | t-6h | t | t+6h | t+8h | experiment id | F6 | F6_r0 | F6_ar4 | F6_r4 | F8 | F8_r0 | F8_ar3 | F8_r6 |
|---|---|---|---|---|---|---|---|---|---|---|---|---|---|---|---|
| | | | | | | | value | | | number of I/O dimensions | | | | | |
| Input — Discharge | +6h Station | | Q[t-6h] | Q[t] | | | Q[t]-Q[t-6h] | 5 | 5 | 5 | 5 | | | | |
| | +8h Station | Q[t-8h] | | Q[t] | | | Q[t]-Q[t-8h] | | | | | 6 | 6 | 6 | 6 |
| Input — Rainfall | instantaneous | | | r[t] | | | r[t] | 1 | | | | 1 | | | |
| | vh average | | avg(r[t-vh:t]) | | | | avg(r[t-vh:t]) | | 1 | | | | | | |
| | vh time-serie | | r[t-vh:t] | | | | r[t-vh:t] | | | v | | | | | v |
| Input — Baseline — FFS' model | | | | Q[t] | Q[t+6h] | | Q[t+6h]-Q[t] | +1 | +1 | +1 | +1 | | | | |
| Input — Baseline — Linear Regression | | | | Q[t] | Q[t+6h] | | Q[t+6h]-Q[t] | +1 | +1 | +1 | +1 | | | | |
| Input — Baseline — Linear Regression | | | | Q[t] | | Q[t+8h] | Q[t+8h]-Q[t] | | | | | +1 | +1 | +1 | +1 |
| | Total Inputs Z | | | | | | | 5/7 | 6/8 | 6/8 | 5+v/7+v | 6/7 | 7/8 | 7/8 | 6+v/7+v |
| Output — Discharge — Toulouse | | | | Q[t] | Q[t+6h] | | Q[t+6h]-Q[t] | 1 | 1 | 1 | 1 | | | | |
| | | | | Q[t] | | Q[t+8h] | Q[t+8h]-Q[t] | | | | | 1 | 1 | 1 | 1 |
| | Total Outputs | | | | | | | 1 | 1 | 1 | 1 | 1 | 1 | 1 | 1 |

The DB is split into a learning dataset (LDS) and a testing dataset (TDS) with a $80\%-20\%$ ratio with two different strategies. The most commonly used strategy in ML for temporal DBs is chronological (denoted by subscript $C$). Its singles out the most recent events for testing. In a scarce data layout, this can infer a bias in TDS. A deterministic (user-defined) setting, noted $D$, is thus proposed. It allows to take the heterogeneity in the data into account and study the impact of extreme events data in the learning process. The 36 events are classified in Table 2 by degrees of severity (from green to orange with respect to FFS warning levels) and types of weather influence. The asterisk indicates the presence of a multiple peak event for that class. Within each class, the number of events in TDS is indicated as a superscript (resp. subscript) for deterministic (resp. chronological) settings. It should be noted that flood events may also differ due to rainfall spatial distribution, snow melt from the Pyrenees and seasonality as detailed in Appendix B. The characteristics of the events in $TDS_D$ are presented in Table 3. For

**Table 2.** Number of events classified by degrees of severity and type of weather influence. Asterisk indicate double peak events. The number of events in TDS is indicated as a superscript (resp. subscript) for Deterministic (resp. Chronological) settings.

| Weather Influence \ Warning level | Green | Yellow | Orange | Total |
|---|---|---|---|---|
| North-North West | 11* [1/2] | 5* [1/1] | 2 [1/1] | 18 [3/4] |
| South-South West | 4* [/1] | 3* [1/] | 0 | 7 [1/1] |
| Eastern Residuals | 3* [1/1] | 2* | 0 | 5 [1/1] |
| South then Eastern Residuals | 2 [1/] | 1 | 0 | 3 [1/] |
| West | 3* [1/1] | 0 | 0 | 3 [1/1] |
| Total | 23 [4/5] | 11 [2/1] | 2 [1/1] | 36 |

each event, it indicates the date, the severity with respect to warning level, the weather influence, the number of peaks $N_{peak}$, the maximum water height in TPN $max(h_{tpn})$, the total rainfall $sum(r)$ and the maximum rainfall $max(r)$. $TDS_D$ is devised to statistically represent the DB in terms of warning levels and weather influences, while maximizing the other characteristics. $TDS_D$ includes 3 events with North-North-Western influence (predominant influence), two of which (27/02/2015, 11/01/2022)





are the biggest events over the entire period with orange and yellow warning levels, and a smaller event on 12/12/2020. $\text{TDS}_D$ also includes 2 multiple-peak events (17/04/2007 and 27/02/2015), Eastern, South-Eastern and Western influenced events (17/04/2007, 24/01/2009 and 22/05/2012). It should be noted that the outlier event (16/07/2018, characterized by heavy rain) is included in $\text{LDS}_C$ while it is included in $\text{TDS}_D$. This allows to diagnose the capability of the learning strategy to predict significant events that are not in LDS. The number of available time-steps $N_\tau$ for the full DB for both $\tau6$ and $\tau8$ forecast model is recalled in Table 4. $N_\tau$ multiplied by the number of input channels $Z$ gives the size of the DB, as shown in 2.2. The number of available time-steps is also given for LDS and TDS with the chronological and deterministic strategies.

**Table 3.** Characteristics of the events included in the testing dataset for the Deterministic setting $\text{TDS}_D$. The events are dated and presented along with their warning level, weather influence, the number of peaks ($N_{peak}$), the maximum water height ($max(h_{tpn})$), the total rainfall ($sum(r)$) and the maximum rainfall ($max(r)$)

| Date | Warning level | Weather influence | $N_{peak}$ | $max(h_{\text{tpn}})$ | $sum(r)$ | $max(r)$ |
|---|---|---|---|---|---|---|
| 17/04/2007 | Green | Eastern residuals | 2 | 2.02 m | 30.5 mm | 5.1 mm |
| 24/01/2009 | Green | West | 1 | 2.37 m | 75.1 mm | 4.7 mm |
| 22/05/2012 | Green | South then Eastern residuals | 1 | 2.02 m | 44.1 mm | 2.7 mm |
| 27/02/2015 | Yellow | North-North West | 3 | 2.92 m | 62.1 mm | 2.6 mm |
| 16/07/2018 | Yellow | South-South West | 1 | 2.56 m | 92.9 mm | 14.9 mm |
| 12/12/2020 | Green | North-North West | 1 | 2.21 m | 62.0 mm | 4.3 mm |
| 11/01/2022 | Orange | North-North West | 1 | 4.31 m | 77.0 mm | 4.3 mm |

**Table 4.** Number of time-steps in LDS and TDS with chronological (C) and deterministic (D) strategies, for $\tau6$ and $\tau8$ forecast models.

| database | number of time-steps in the DB |
|---|---|
| $\tau6$ | $N_{\tau6} = 4404$ |
| $\tau8$ | $N_{\tau8} = 4002$ |
| $\tau6$, TDS, C | $N_{\tau6,\text{TDS},C} = 1003$ |
| $\tau6$, LDS, C | $N_{\tau6,\text{LDS},C} = 3401$ |
| $\tau6$, TDS, D | $N_{\tau6,\text{TDS},D} = 955$ |
| $\tau6$, LDS, D | $N_{\tau6,\text{LDS},D} = 3449$ |
| $\tau8$, TDS, C | $N_{\tau8,\text{TDS},C} = 975$ |
| $\tau8$, LDS, C | $N_{\tau8,\text{LDS},C} = 3027$ |
| $\tau8$, TDS, D | $N_{\tau8,\text{TDS},D} = 750$ |
| $\tau8$, LDS, D | $N_{\tau8,\text{LDS},D} = 3252$ |





## 3 Data-driven strategies for flood forecasting

Three ML models are used to forecast the discharge of the Garonne river at TPN using inputs described in Sect. 2: Linear Regression (LR), Gradient Boosting Regressor (GBR), and Multilayer Perceptron (MLP). These algorithms are detailed in

Sect. 3.1. The hyper-parameters for these algorithms are set with a Cross Validation (CV) step described in Sect. 3.2. The performance of the trained models is then assessed with appropriate metrics and criteria in Sect. 3.3.

In the following, the size of the DB is noted $N_{tot}$. The number of time-steps (amongst all the events) is noted $N_\tau$, declined as $N_{\tau,\text{LDS}}$ and $N_{\tau,\text{TDS}}$ for LDS and TDS, respectively. The input data of the studied events is a second order tensor noted $\boldsymbol{X} \in \mathbb{R}^{Z \times N_\tau}$ where $Z$ is the number of input channels. The ML models are trained to predict an ensemble of discharge values

in TPN, issued hourly at the targeted forecast lead time, for all events in the DB. These discharge values are arranged as a vector (first order tensor): $\boldsymbol{y} = (y_1 \cdots y_t \cdots y_{N_\tau})$ with $y_t = \Delta Q_{tpn}(t) \in \mathbb{R}$ and $t \in [\![1, N_\tau]\!]$. The learning models $f$ provide the prediction $\widetilde{y}_t = f(\boldsymbol{x}_t)$ associated to a realization of the input $\boldsymbol{x}_t = (x_{1,t} \cdots x_{i,t} \cdots x_{Z,t}) \in \mathbb{R}^Z$, where $\forall t \in [\![1, N_\tau]\!]$,

$$
x_{i,t} = \begin{cases} \Delta Q_i(t-\tau), & \text{for } i \in \{Net\} \\ F_{rain}(r(t), \nu) & \text{for } i = \nu \end{cases}
$$

with the set $\{Net\}$ an ensemble of $\tau6$ or $\tau8$ network locations and $\nu \in [\![-24, -2]\!] \cap 0 \cap [\![2, 24]\!]$, as specified in eq. 2.2. Their ten-

sor counterparts, $\boldsymbol{f}$, provide the prediction $\widetilde{\boldsymbol{y}}$ associated to the entire set of studied events with inputs $\boldsymbol{X} = (\boldsymbol{x}_1^T \cdots \boldsymbol{x}_t^T \cdots \boldsymbol{x}_N^T) \in \mathbb{R}^{Z \times N_\tau}$. These notations are summarized in Table 5.

**Table 5.** Input-Output notations used in this study. $Z$ is the number of input channels. $N_\tau$ is the number of time-steps.

| variable | single time-step | $N_\tau$ time-steps |
|---|---|---|
| input | $\boldsymbol{x}_t$ | $\boldsymbol{X}$ |
| dimensions | $Z$ | $Z \times N_\tau$ |
| Output | $y_t$ | $\boldsymbol{y}$ |
| Model function | $f(\boldsymbol{x}_t) = \widetilde{y}_t$ | $\boldsymbol{f}(\boldsymbol{X}) = \widetilde{\boldsymbol{y}}$ |
| dimensions | 1 | $N_\tau$ |

### 3.1 Standard Machine Learning algorithms

#### 3.1.1 Linear Regression

For a given time $t$, the Linear Regression identifies a linear combination of the elements in a realization of the inputs $\boldsymbol{x}_t$ in

LDS (or TDS), to represent the output $y_t$. The linear regression function $f_{lr}$ reads:

$$
\forall t \in [\![1, N_{\text{LDS}}]\!], \quad \widetilde{y}_t = f_{\text{lr}}(\boldsymbol{x}_t) = \alpha_0 + \sum_{i=1}^{Z} \alpha_i x_{i,t} = \alpha_0 + \boldsymbol{\alpha}\boldsymbol{x}_t^T \tag{3.1}
$$



with $\boldsymbol{\alpha} = (\alpha_1 \cdots \alpha_i \cdots \alpha_Z)$ and $\alpha_i \in \mathbb{R}, \forall i \in [\![1, Z]\!]$. This generalizes to a full set (e.g. LDS) gathering all time steps for all events:

$$\widetilde{\boldsymbol{y}}_{\text{LDS}} = \boldsymbol{f}_{lr}(\boldsymbol{X}_{\text{LDS}}) = \boldsymbol{\alpha}_0 + \boldsymbol{\alpha}\boldsymbol{X}_{\text{LDS}}. \tag{3.2}$$

This problem can be described as finding the vector $\boldsymbol{\alpha}$ of $Z$ coefficients $\alpha_i$ that best approximates $\boldsymbol{y}_{\text{LDS}}$ with $\widetilde{\boldsymbol{y}}_{\text{LDS}}$. This is achieved solving the optimization problem considering the $L_2$ norm $\min_{\boldsymbol{\alpha}} \|\boldsymbol{\alpha}\boldsymbol{X}_{\text{LDS}} - \boldsymbol{y}_{\text{LDS}}\|_2$. Here, the input features in $\boldsymbol{x}_t$ are supposed independent or weakly correlated to avoid multicollinearity effects.

### 3.1.2 Gradient Boosting Regressor

GBR is an ML technique for regression problems. It relies on the recursive correction of the loss of weak estimators, which
commonly are decision trees (DTs) (or more precisely regression trees (RTs) in this study). The first algorithm based on boosting was called Adaboost (Freund and Schapire, 1995; Freund et al., 1996), and proven to outperform CART-like algorithms (proposed by Breiman et al. (1984)) in terms of generalization. The concept was also investigated by Breiman (1997) (classification) and extended by Friedman (1999, 2001) (classification and regression). The latter was implemented in the Scikit-Learn library (Pedregossa et al., 2011) used in this paper.

DTs (Montalbano, 1962; Pollack, 1965) allow to efficiently make decision from tables of data, setting splitting rules recursively upon the inputs. Branching inputs results in categorized outputs, called leaves, organized in a *classification tree* model. Some leaves can be pruned from the result; relieving the tree from the least prominent branches and allowing for a better generalization outside LDS. As presented in Alg. 1, RTs share the core building logic of DTs, but differ in a few subtle ways. The splitting strategy focuses on minimizing an MSE, instead of losses that fit classification problems. Also, the output is
interpreted as a real-valued variable, instead of a probability of belonging to a discrete class. The learning process is optimized by a CART algorithm originally theorized by Breiman et al. (1984).

RTs require little data processing (as they can handle categorical and numerical data), are rather simple to understand (work like human decisions), and can handle co-linearities (e.g. appears with boosting). RTs are also known to be very sensitive to changes in the data and can easily over-fit on the training data.

---

**Algorithm 1** Regression tree model inference process

---

**Require:** TDS or LDS, binning of the input space, maximum tree depth $M$

   **for** $m \in [\![1, M]\!]$ **do**

      Split of TDS's inputs according to predictor and splitting criteria (set during optimization on LDS).

   **end for**

   Each final split is called a leaf and its value is based on the average value of the predictor (set during optimization).

---

The Gradient Boosting Regressor model is presented in Alg. 2. It relies on correcting the residuals from the recursive addition of the outputs of multiple weak estimators, such as RTs. This special type of ensemble method yields strong models, at the expenses of reduced interpretability and higher computational cost.



---

**Algorithm 2** Gradient Boosting Regressor model inference process

---

**Require:** TDS, $B$ boosting stages with trees of depth $M$.

**Require:** Predict output residuals from TDS inputs with a RT of depth $M$. (with possible pruning)

   The result, identified as Regressor, is the sum of the outputs of the RTs

   **for** $b \in [\![1, B]\!]$ **do**

       Predict Regressor residuals from TDS inputs with a new RT of depth $M$ (with possible pruning)

       Sum the new RT's and the Regressor's outputs.

   **end for**

   The Regressor is the output of the last iteration.

---

   The GBR has a variety of parameters to set and/or calibrate. Some of them stem from the RT such as the tree depth or pruning threshold. The default values of the scikit-learn package proposed by Pedregossa et al. (2011) were kept for most available RT
parameters, such as the total number of leaves, which is the possible number of output values. Other parameters, optimized (further details in Sect. 3.2) to bolster the learning phase, are related to the boosting process, such as the number of estimators, the learning rate (shrinkage of the tree's contributions) and early stopping parameters. The loss function used to monitor the boosting algorithm is a Huber function (Friedman, 2001), more robust to outliers, instead of the default squared error. The data is not standardized as it is not required for this kind of model.

### 3.1.3   Multilayer Perceptron

Multilayer Perceptrons take their origins in the 1940s with McCulloch and Pitts (1943) showing the ability to compute logical functions and model a neuron's behavior through mathematical representations of Neural Networks. As shown in Eq. (3.3), a neuron is modeled through a linear combination of the inputs, shifted with a bias (making the linear combination affine), then fed to a chosen non-linear function (an activation function $g$, e.g. sigmoid, hyperbolic tangent, ReLu, etc...). The coefficients
of this "artificial neuron": $\boldsymbol{\alpha} = (\alpha_1 \cdots \alpha_i \cdots \alpha_Z)$ of the linear combination (called weights) and the bias $\alpha_0$ are fitted to a given LDS:

$$\forall t \in [\![1, N_{\mathrm{LDS}}]\!], \quad \widetilde{y}_t = f_{\mathrm{neuron}}(\boldsymbol{x}_t) = g(\alpha_0 + \sum_{i=1}^{Z} \alpha_i x_{i,t}) = g(\alpha_0 + \boldsymbol{\alpha}\boldsymbol{x}_t^T). \tag{3.3}$$

   In the late 1950s, during the early stages of the Artificial Intelligence (AI) field, Rosenblatt (1958) represented cognitive
perception with a network composed of multiple layers of the aforementioned artificial neurons, called Perceptron. For fully connected layers, every neuron of a layer computes an output according to its own weights by taking the outputs from the previous layer as its inputs. A MLP model as described in Alg. 3 is a fully connected NN. Single-layer networks can only represent limited phenomena, as shown by Minsky and Papert (1969) while multi-layer networks allow to represent complex phenomena. MLP is a simple feed-forward NN, and is trained using a backpropagation algorithm (e.g.:RMSProp, Adam, see
chap. 8 of Goodfellow et al. (2016) for more). LDS is used to assess the coefficients of the neurons (internal to the model),



---

**Algorithm 3** Multilayer Perceptron inference process.

---

**Require:** $H$ layers of size $\{M_1, \cdots M_h, \cdots M_H\}$.

The input layer: A layer composed of $M_1$ neurons $f_{\text{neuron}}$ each with their own weight values.

With the same $Z$ inputs, each neuron computes an output according to its weights.

**for** $h \in [\![2, M_H]\!]$ **do**

    A layer composed of $M_h$ neurons $f_{\text{neuron}}$ each having separate weight values.

    **for** $j \in [\![1, h]\!]$ **do**

        The $j^{th}$ neuron computes an output with every output of the previous layer's neurons as its inputs.

    **end for**

**end for**

The output (last) layer: a single neuron $f_{\text{neuron}}$. With every output of the last hidden layer's neurons, it computes an output for the model.

This layer is usually not bound between 0 and 1.

---

then the gradient of a loss function is back-propagated through the network to fit the coefficients; this process iterates until the convergence criteria on LDS are met (for generalization purposes). To monitor and control this internal process, some global (hyper-)parameters of the MLP are set or optimized (see sect. 3.2).

In the present work, at neuron-level, the activation function is set to ReLu, as it is known to avoid vanishing gradients of the
solver's loss function during its back-propagation through the network (training phase). At network-level, the size of the layers and the number of neurons are optimized. The number of epochs and the learning rate are actively adapted during training, in complement with early stopping parameters that are also used. The solver used in this paper is presented by Adam (Kingma and Ba, 2017); it features a stochastic gradient-descent method, with its associated parameters that should be set. The Scikit-learn (Pedregossa et al., 2011) MLP implementation was here favored for its simplicity, usability and compatibility with other
ML models. However, for larger MLP models, using bigger DBs, an implementation with Keras (and others Chollet, 2015) or PyTorch (Paszke et al., 2019) should be favored as these solutions are more adapted and compatible with GPUs and TPUs.

### 3.2 Hyper-parameter optimization

*Grid Searching* consists in the optimization of the hyper-parameters of the ML algorithms to improve the robustness of the solution. The optimization is achieved on a subset of LDS while TDS is used for final validation only. This procedure decreases
the size of the learning data set, thus introducing more biases. A *Grouped K-fold Cross Validation* is implemented to limit the introduction of a bias that may result from the reduction of the learning data set. LDS is divided in 5 folds, 4 of them are used for training while the last fold is used for validation. The optimization is carried out for each selection of 4 of the 5 folds; the remaining one is used for validation. The average of the validation scores are used to evaluate the performances of each set of hyper-parameters and select the best set. This computationally expensive strategy is applicable in the present case, given the
small size of the DB. In the present study, LR, GBR and MLP algorithms from scikit-learn Pedregossa et al. (2011) are used.





Two hyper-parameters are used for GBR (number of boosting stages and pruning criterion) and MLP (number of hidden layers and size of each layer).

### 3.3 Performance metrics

During training, each algorithm uses a dedicated metric to learn and optimize its performance over the training set. The
generalization performance of the models is assessed with three criteria computed on LDS and on TDS after the optimization process, either per set or per event.

#### 3.3.1 Nash-Sutcliffe Efficiency coefficient

The Nash-Sutcliffe Efficiency coefficient (NSE) is a $R^2$ score that assesses the predictive skill of hydrological models. It formulates the ratio between the error variance of the model and the variance of the observation. Values of the NSE close to
1 means high predictive skill. Here, the observation is the observed discharge in TPN. It constitutes the output $y_t$ in LDS and TDS:

$$\text{NSE}(\boldsymbol{y}, \boldsymbol{f}(\boldsymbol{X})) = R^2(\boldsymbol{y}, \boldsymbol{f}(\boldsymbol{X})) = 1 - \frac{\sum_{t=1}^{N_\tau}(y_t - f(\boldsymbol{x}_t))^2}{\sum_{t=1}^{N_\tau}(y_t - \overline{\boldsymbol{y}})^2} \tag{3.4}$$

where $\overline{\boldsymbol{y}} = 1/N_\tau \sum_{t=1}^{N_\tau} y_t$ is the mean of the output values, $n$ is either the length of an event or the sum of the lengths of the events within a data set. NSE is equal to 1 (perfect prediction) for a perfect model with an estimated error variance equal to
zero. NSE=0 means that the model has the same predictive skill as the mean of the observations, in terms of squared error, meaning that the model produces an estimated error variance equal to the variance of the observations. When the estimated error variance of the model is significantly larger than the variance of the observations, the observed mean is a better predictor than the model and NSE < 0.

#### 3.3.2 Persistence criterion

The persistence criterion $P$ assesses whether the model is able to be more accurate than a persistent model (repetition of the input as output). In this case, this means that it repeats the input flow in TPN with a fixed lag of $\tau6$ or $\tau8$:

$$P = (\boldsymbol{y}, \boldsymbol{f}(\boldsymbol{X})) = 1 - \frac{\sum_{t=\tau}^{N_\tau}(f(\boldsymbol{x}_t) - y_t)^2}{\sum_{t=\tau}^{N_\tau}(y_{t-\tau} - y_t)^2} \tag{3.5}$$

where $\tau$ is the predetermined shift of the time series. $P$ is computed by event and the value per set is the mean computed over the events. Persitence values close to 1 suggest that the prediction is more accurate then the persistence model. $P = 0$ means
that the model has the same predictive skill as a persistent model, and $P \leq 0$ show an even more offset prediction of the model.

#### 3.3.3 Relative Peak Error

The Relative Peak Error (RPE) Song et al. (2020) evaluates the quality of the model at the flood peak with respect to the observed discharge, which is of great importance for risk assessment. The RPE per event is computed for time steps that




present water height in TPN that exceeds the alert threshold $H_t = 1.70m$ prescibed by FFS. The RPE per set is the average
over the events. The RPE reads:

$$\text{RPE}(\boldsymbol{y}, \boldsymbol{f}(\boldsymbol{X})) = \frac{1}{n_{\mathcal{P}}} \sum_{t \in \mathcal{P}} \frac{f(\boldsymbol{x}_t) - y_t}{y_t} \tag{3.6}$$

where $\mathcal{P} = \{t = arg\,\underset{t}{max}(y_t), y_t \in \mathcal{R}_{event}\}$ with $\mathcal{R}_{event}$ the subset of output values $y_t$ above $H_t$. Values closer to zero indicate
a good estimation of peak discharges.

## 4 Results

### 4.1 Learning with discharge data

This section presents the results from ML models for $\tau 6$ and $\tau 8$ using input data described in Table 1.

Figure 2 presents the persistence score along the x-axis, for F6 and F8 described in Table 1, using the chronological split
noted $C$. The different algorithm are listed along the y-axis. Results for F6 (resp. F8) are indicated in the white (resp. gray)
boxes. Note that only F6 is available for the FFS ELR model. Figure 2a provides the persistence computed over all the events
in the DB (either LDS or TDS) while Fig. 2b provides the persistence computed by event. The score is indicated by a colored
dot, for F6 (resp. F8): orange (resp. red) is used on LDS while light (resp. dark) green is used on TDS. A green line indicates
that the score is better in learning than in test, which is expected. On the other hand, an orange or red line indicates that
the score is better in test, which is unexpected and should raise awareness. Two specific events are highlighted on panel (b):
the upward blue triangle indicates the January 2022 event that presents the highest peak discharge (and highest peak water
height $max(h_{\text{tpn}})$ of the DB and the downward red triangle indicates the July 2018 event that presents the highest peak rainfall
$max(r)$. Figure 2a shows that all ML models outperform ELR for F6,C for training on LDS with $P$ scores beyond 0.9 (against

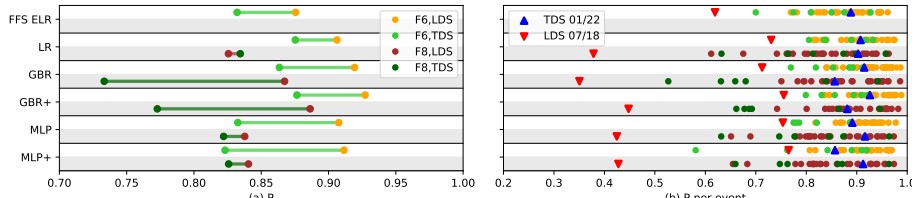

**Figure 2.** Persistence scores $P$ on all models for sets and events in the F6,C and F8,C settings. The criteria are computed by sets in (a) and
by events in (b). Orange (resp. red) dots are for LDS events in F6,C (resp. F8,C) and light (resp. dark) green dots for TDS events. On the
right-hand panel, the blue upward triangle indicates the Jan. 2022 20-year event. It is included in TDS. The downward red triangle indicates
the Jul. 2018 event with the biggest instantaneous rain. It is included in LDS.

$P = 0.87$ for ELR). The best score is obtained with GBR and GBR+. Only MLP and MLP+ present a slightly smaller score
than ELR. For a lead time of $\tau 6$, as expected, for all models, TDS scores are smaller than those of LDS. LR and GBR(+) scores
remain higher than those of ELR for TDS. LR and GBR(+) models show similar results ($P \approx 0.87$) with higher score than



ELR ($P = 0.83$). However, MLP ($P = 0.82$) underperforms with respect to ELR, even with the addition of information from the baseline in MLP/+'s. This is most likely due to a slight over-fit of the MLP/+ models to LDS, as shown in Fig. 2b, where all or most of TDS events score poorly. When the lead time is extended to $\tau 8$, the performance of all models decreases due to the complexity of the dynamics over the upstream catchment. As expected, the scores are higher for LDS than for TDS, except for LR which points to the limitations of the choice made here for splitting the DB with the chronological setting. GBR(+)

provides the best scores in learning ($P_{GBR} = 0.86$ and $P_{GBR+} = 0.88$) while it provides the worst score in test ($P_{GBR} = 0.73$ and $P_{GBR+} = 0.77$). Additional scores are shown in Fig. C1).

The results shown in Fig. 2b highlights the merits and the limitations of the chronological setting for the scarce DB using discharge data only. Blue triangles indicates scores for the January 2022 event in TDS. This event is the most important event with a 20-year return period present in DB, for $\tau 6$ and $\tau 8$. All ML models score well on this event, suggesting that it is well

described by upstream discharge provided at the selected stations for F6 and F8 and that it is weakly influenced by rainfall over the intermediate catchment. Yet, the event in July 2018 (indicated with red triangles) in LDS is poorly learned for both F6 and F8, with all ML models. This event is characterized by heavy instantaneous rainfall on the intermediary catchment. Including this complex event in LDS leads to poor results in learning as rainfall is not included in the input data. The observed water height in TPN is plotted as a time series in Fig. 3 for July 2018 event (thick black curve, left y-axis). The hyetogram is plotted

in blue (reversed right y-axis) and shows that the maximum water height is observed a few hours after the maximum rainfall. The water heights predicted by the models are plotted with different colors. It appears that all model fail to forecast the peak, as the maximum water height is lately predicted with a 6-hour time lag. It should be noted that there is no event similar to that of July 2018 available in the DB. Thus the events in TDS are easier to predict, leading to a slightly better LR score for F8,TDS than F8,LDS (Fig. 2b). While ML models, tend to better learn this event, they provide poor score on TDS, i.e. for F6 with

MLP/+. This advocates for the extension of the input data to rainfall, especially when increasing the forecast lead time. This also questions the split of events for the chronological setting and advocates for an alternative deterministic setting denoted by the subscript $D$. The event of July 2018 being unique in the DB, it is best used in TDS for the deterministic setting to assess the generalization abilities of the models for rare and intense rainfall-based events. Thus, F6,D and F8,D settings are used in the following.

Similarly to Fig. 2, Fig. 4, presents the persistence score for F6 and F8 for all models, using the deterministic setting $D$ (July 2018 event is now included in TDS). With this setting, and for the lead time $\tau 6$, ML models outperform ELR. The ML models all present better results in learning than in test, as expected. TDS performance in test are similar for all models (except that of GBR,F8), showing that the previously highlighted sensitivity to extreme events has been removed. July 2018 remains poorly predicted with the deterministic setting (not shown). Thus, in spite of the global improvement brought by the deterministic

setting, the extension of the input data to rain fall data remains necessary.





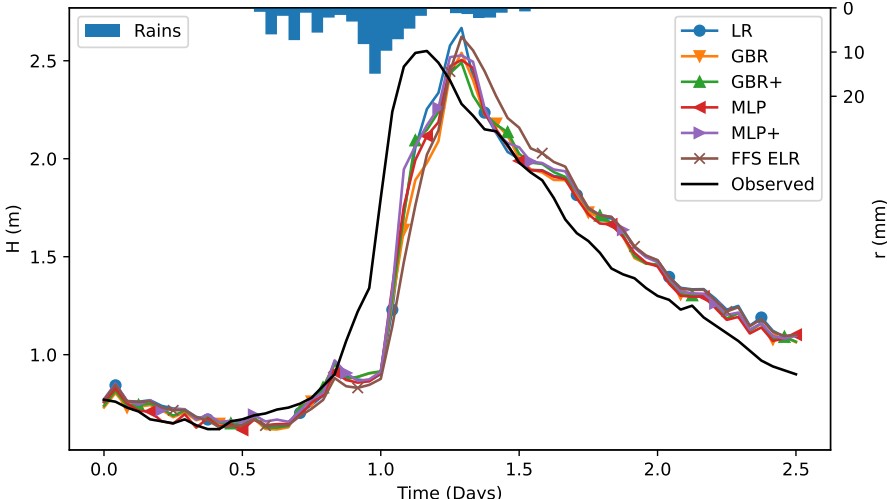

**Figure 3.** Water height for July 2028 event included in LDS for F6,C. The observation $h_{tpn}$ is plotted with a thick black line, the water height learned from ML models are plotted in color. The observed hyetogram $r$ is plotted at the top of the panel and reversed right y-axis.

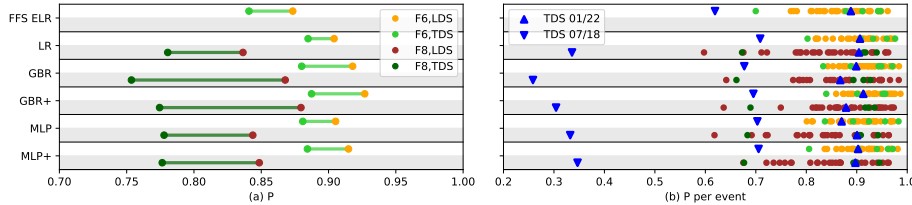

**Figure 4.** Persistence scores $P$ for all models for F6,D and F8,D settings. The criteria is computed by sets in (a) and by events in (b). Orange (resp. red) dots are for LDS events in F6,D (resp. F8,D) and light (resp. dark) green dots for TDS events. On the right panel, the blue upward triangle indicates the Jan. 2022 20-year event. It is included in TDS. The downward red triangle indicates the Jul. 2018 event with the biggest instantaneous rain. It is included in TDS.

## 4.2 Improving forecasts with rainfall data

### 4.2.1 +6 h lead time

The addition of rainfall data to the prediction is first assessed at $\tau 6$. As mentioned in Sect. 2.2, this data is used either instantaneously ($F6\_r0$), time-averaged over $\nu$ h ($F6\_ar\nu$) or as a collection of data over $\nu$ h. For each algorithm, the time interval
$\nu$ h is chosen to maximize NSE and $P$ while keeping RPE close to 0. More details are provided in appendix D.

  Figure 5 presents the models persistence $P$ for the 6 h lead time ($\tau 6$) for the three rainfall data configuration $F6\_r0$, $F6\_ar\nu$ and $F6\_r\nu$, as well as $F6$ without rainfall data. Figure 5a presents the results computed per set while Fig. 5b presents the results per event. LDS (reps. TDS) scores are indicated with green (resp. yellow) symbols and each rainfall setting is





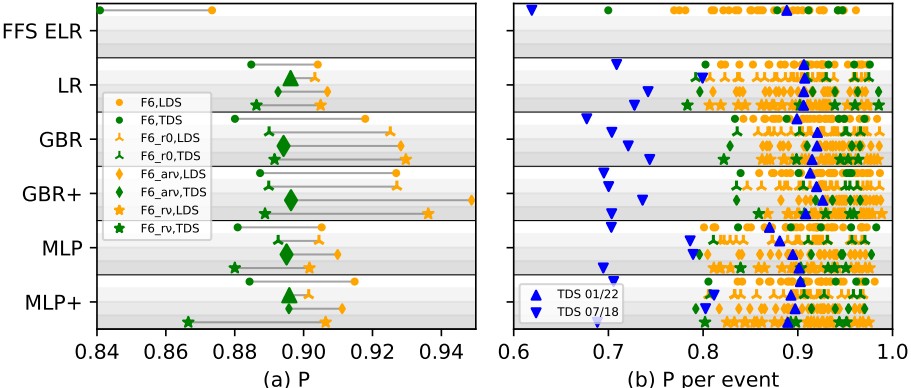

**Figure 5.** Persistence for all models for configurations $F6$, $F6\_r0$, $F6\_ar\nu$ and $F6\_r\nu$ (resp. dot, triangle, diamond, star). Left (resp. right) panel: $P$ computed per set (resp. event). Orange (resp. green) symbols indicate $P$ for LDS (resp. TDS). The Jan. 2022 event is indicated as a blue upward triangle and the Jul. 2018 is indicated as a downward blue triangle. Both events are in TDS).

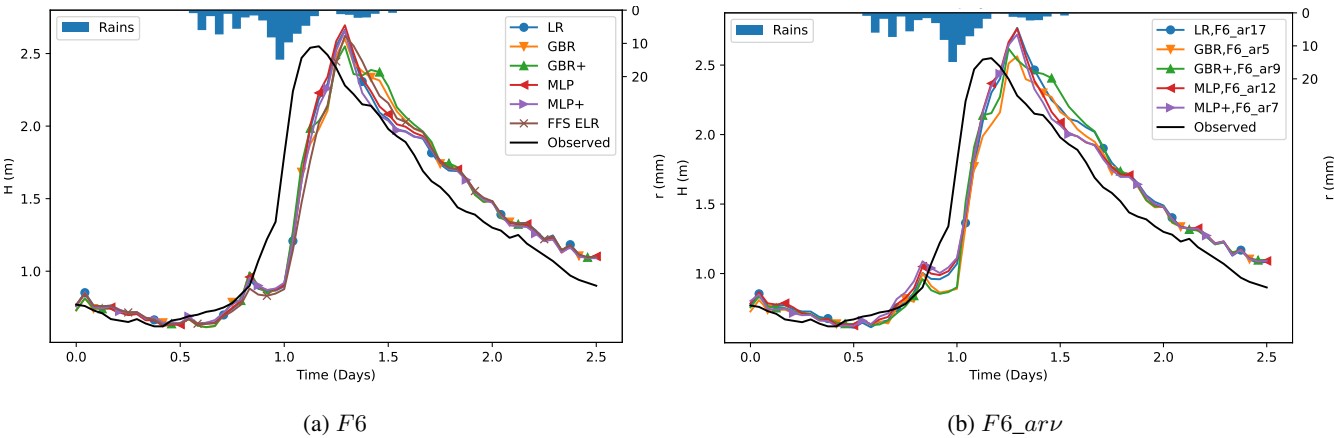

**Figure 6.** Water height for July 2018 event in TDS for $F6$ and $F6\_ar\nu$. $h_{tpn}$ is plotted with a thick black line, the water height learned from ML models are plotted in color. The observed hyetogram $r$ is plotted at the top of the panel and along the reversed right y-axis.

indicated with a different symbol. The NSE and RPE criteria are shown in Fig. C3 in App. C with similar layout. Figure 5a
indicates that including rainfall data improves TDS performance of LR and GBR/+ models, for all rainfall data configuration. Yet, the addition of rainfall data only improves the performance of MLP/+ when taken instantaneously or time-averaged. It seems that potentially highly correlated entries in $F6\_r\nu$ configuration may disrupt the learning process in MLP/+. Figure 5 also illustrates the robustness of the results. Indeed, all algorithms, applied with their own optimal rainfall data pre-processing, provide similar results for this scarce data case study. The July 2018 outlier remains the most poorly predicted event, in spite of
the addition of rainfall data. Yet the addition of rain data lead to slight improvement with respect to the no-rain configuration. It should be noted that for $F6\_r\nu$ results, MLP/+ has the lowest prediction scores. This is most likely due to over-fitting.



Figure 6 shows water height predictions for the July 2018 event for configurations $F6$ (without rains, Fig. 6a) and $F6\_ar\nu$ (averaged rains, Fig. 6b). $h_{tpn}$ is represented with a thick black line, the water heights predicted from ML algorithms are plotted in color. The observed hyetogram $r$ is plotted at the top of the panel and along the reversed right y-axis. It appears that the

prediction is not improved by the addition of time-averaged rainfall data, and that the flood peak remains unpredicted. MLP/+ and LR with rainfall averaging ($F6\_ar\nu$ setting) bring a slight improvement at the beginning of the event. Yet, the July 2018 remains an outlier for the learning given the scarcity of the DB.

#### 4.2.2    +8 h lead time

The addition of rainfall data to the prediction is here assessed at $\tau8$ with settings $F8\_r0$, $F8\_ar\nu$ and $F8\_r\nu$. The rainfall data selection process, already used in sect. 4.2.1 for $\tau6$, is explained in appendix D for $\tau8$.

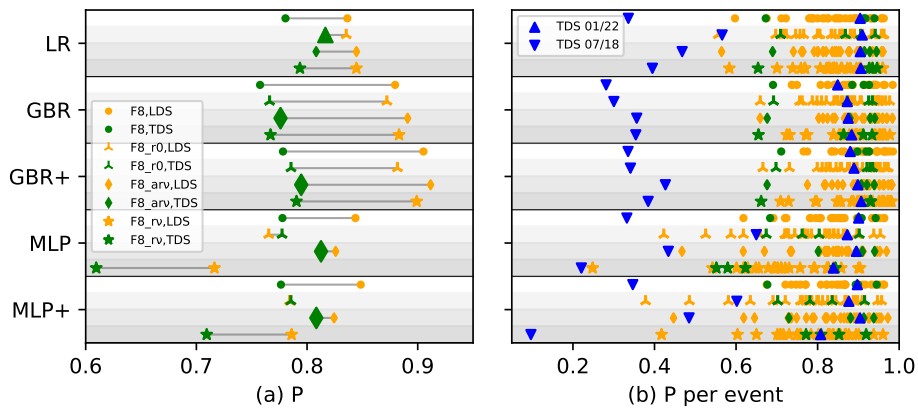

**Figure 7.** Persistence for all models for configurations $F8$, $F8\_r0$, $F8\_ar\nu$ and $F8\_r\nu$ (resp. dot, triangle, diamond, star). Left (resp. right) panel: $P$ computed per set (resp. event). Orange (resp. green) symbols indicate $P$ for LDS (resp. TDS). The Jan. 2022 event is indicated as a blue upward triangle and the Jul. 2018 is indicated as a downward blue triangle. Both events are in TDS).


Figure 7 presents the models persistence $P$ for $\tau8$ for the three rainfall data configuration $F8\_r0$, $F8\_ar\nu$ and $F8\_r\nu$, as well as $F8$ without rain fall. The figure layout is similar to that of Fig. 5. NSE and RPE scores are illustrated in App. C.

Figure 7a illustrates that including rainfall data slightly improves TDS performance of the LR and GBR/+ models. Yet, the performance is sensitive to the rainfall data setting. Indeed the instantaneous setting $F8\_r0$ is best for GBR while the averaged

data setting $F8\_ar\nu$ performs best for LR and GBR+. $F8\_ar\nu$ also appears to be the most favorable configuration for MLP/+. For all algorithms, the addition of rainfall data as time-series does not bring improvement, it even severely worsens the results for MLP/+. Similarly to conclusions at $\tau6$, Fig. 7a and Fig. 7b illustrate the robustness of our results as all algorithms, applied with their own optimal rainfall data pre-processing, provide similar results. It appears that averaging the rainfall data over a time interval allows to reduce the variability of $P$ scores per event (Fig. 7). We here conclude that rainfall data averaging is

the best strategy to take these additional data into account in the learning process. As previously noted at $\tau6$, MLP/+ presents higher sensitivity to data settings. It tends to over-fit the rainfall data with $F6\_r\nu$ setting and provides low score on TDS.





While GBR is more robust to correlated inputs and provides good scores (it also learns best with LDS), LR performs slightly better in TDS. LR here appears as the most efficient model for this case study, given the small DB size and the relatively linear process considered here. Yet, the robustness and efficiency of LR should be investigated in the presence of larger non-linearities, complex processes involved in further extended lead-times. Water height prediction for $\tau 8$ are shown in Fig. 8 for

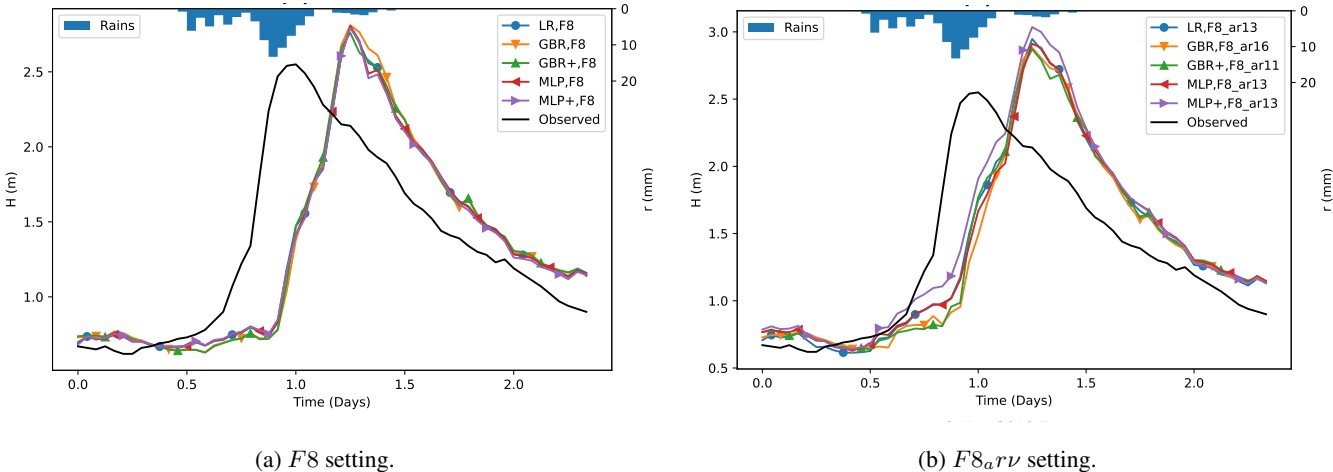

(a) $F8$ setting.

(b) $F8_a r\nu$ setting.

**Figure 8.** Water height for July 2018 event in TDS for $F8$ and $F8\_ar\nu$. $h_{tpn}$ is plotted with a thick black line, the water height learned from ML models are plotted in color. The observed hyetogram $r$ is plotted at the top of the panel and along the reversed right y-axis.

July 2018 event for F8 (without rains, Fig. 8a) and $F8\_ar\nu$ (averaged rains, Fig. 8b) with similar layout as in Fig. 6. The addition of averaged rainfall data tends to improve MLP/+ predictions at the beginning of the event, the flood rise and peak remain unpredicted. Indeed, the ML models $P$ score below $0.5$ for July 2018, which is the lowest score by event. It should be noted that the complexity of the prediction at extended lead time results in low scores for several events.

## 5 Conclusions

This study discusses the potential of ML models on short-term flood forecasts with a scarce and heterogeneous DB, applied to predict discharges in Toulouse from upstream discharges and rainfall information. The empirical Lag and Route (ELR) model from the FFS serves as a reference for $6$ h forecast lead time. The major finding of this work is that learning from only discharge data at in-situ hydrometric stations, ML models outperform the current FFS ELR. Based on these conclusions, the ML strategy was extended to $8$ h forecast. It was established that for longer forecast lead times, additional data should be taken into account. Rainfall data over the intermediate basin between upstream stations and Toulouse were then considered.

The specificity of this research study lies in the small size of the DB, the scarcity of extreme events, some of them originating from different weathers. For that reason, several strategies for the layout of the learning and test sets were investigated and implemented for ML models at $6$ h forecast and $8$ h forecast. A classical chronological 80%-20% split was first used. This split was used for both lead times, without input rainfall data. For $6$ h lead time, all ML models outperform the ELR. GBR/+



provides the best results in learning while LR and GBR/+ provide the best similar results in prediction. For 8 h lead time, GBR/+ provides the best results in learning but under performs in prediction as opposed to LR and MLP/+. It should be noted that the July 2018 outlier was included in the chronological split learning set which limits the generalization abilities of the models and is prone to over-fitting. A deterministic split was then proposed where the outlier of July 2018 was now included

in the test set. This strategy led to better performances over the DB and less variability through the models results. All LR, GBR/+ and MLP models outperforms ELR. All ML models lead to close performances in prediction, and are, as expected, better for shorter lead times. Also as expected, the scores are better in learning than in prediction. This split leads to more robust results than the chronological split. Yet, some events remain poorly predicted with the deterministic setting. Indeed, it was highlighted that, for complex processes such as those driven by rainfall, input information should be added to the data

base. The deterministic split was favored for experiments where rainfall data are taken into account in the DB. The use of rainfall data requires careful processing as these data are provided as time-series with potential correlation in time. Here, the best results were obtained when averaging rainfall over a past period: $F6\_ar\nu$ and $F8\_ar\nu$. For each model, this length $\nu$ was chosen in order to optimize the learning score. This strategy stands as a compromise between using only one instantaneous rain data ($F6\_r0$ and $F8\_r0$) and using the potentially correlated time-series ($F6\_r\nu$ and $F8\_r\nu$). In conclusion, the use of a

scarce DB enlightens the need for heterogeneous data and fine data processing. It was also highlighted that the simplicity of a LR is well adapted to scarse DBs. Finally, it is expected that learning and predicting performance would greatly benefit from the addition of other major, and heterogeneous, events in the DB.

As a perspective, this study could be extended to other catchments, building a global data base, gathering extreme events and enriching the data, as proposed in Kratzert et al. (2019a). Assessing such large-scale modeling approach with respect to the

local approach implemented here would highlight the merits and limitations of a down-scaling strategy, especially for increased forecast lead times. As expected, the extension of the forecast lead time here comes with a decrease in predictive performance, yet, it avoids the construction of a physics-based hydrology model or an ELR for each lead time. In the present work, this was implemented for 8 h lead time and could be applied for further extended lead times. In that perspective, the use of rainfall data should be investigated, making the most of high resolution spatially distributed rain products over a larger catchment.

Nevertheless, this may call for the addition of major events driven by rainfall in the DB. Splitting the learning processes by first learning from discharge then learning from rain fall could also be investigated. Another perspective for this work lies in the choice of the period over which the input discharge data are selected. Indeed, a major hypothesis lies in the choice of an approximate fixed time transfert for discharge from each observing station upstream of Toulouse. In reality, this time transfert probably differs for each upstream station and may also be flow dependent. Working with variable lead times could favor the

use of more flexible and complex predictors such as LSTMs.

*Code and data availability.* Code and data available upon request.





## Appendix A: Hydrometric stations characteristics

Table A1 shows in-situ stations names along some geographic data.

**Table A1.** Hydrometric stations characteristics, grouped by lead times

| station name | River | Area ($km^2$) |
|---|---|---|
| Toulouse Pont Neuf | Garonne | 10133.95 |
| $\tau 6$ | | |
| Mancioux | Garonne | 2810.96 |
| Mas d'Azil | Arize | 220.21 |
| Mazères | Grand Hers | 1376.84 |
| Roquefort sur Garonne | Salat | 1574.95 |
| Saverdun | Ariège | 1813.83 |
| $\tau 8$ | | |
| Aspet | Ger | 95 |
| Foix | Ariège | 1340 |
| Mas d'Azil | Arize | 220.21 |
| Mirepoix | Grand Hers | 640 |
| Saint-Girons | Salat | 1154 |
| Valentine | Garonne | 2230 |

## Appendix B: Splitting Process

Table B1 gathers data about the various events to be able to classify them. Basic hydraulic data is shown here (warning level, number of peaks and maximum height), as well as weather data (which the models do not see).



**Table B1.** Characteristics of the events included in the DB. The events are dated and presented along with their warning level, weather influence, the number of peaks ($N_{peak}$), the maximum water height ($max(h_{tpn})$), the total rainfall ($sum(r)$) and the maximum of rainfall ($max(r)$)

| Date | Warning level | Weather influence | $N_{peak}$ | $max(h_{\text{tpn}})$ | $sum(r)$ | $max(r)$ |
|---|---|---|---|---|---|---|
| 17/04/2007 | Green | Eastern residuals | 2 | 2.02 m | 30.5 mm | 5.1 mm |
| 24/01/2009 | Green | West | 1 | 2.37 m | 75.1 mm | 4.7 mm |
| 21/04/2009 | Yellow | Eastern residuals | 1 | 2.57 m | 38.7 mm | 6.0 mm |
| 01/05/2009 | Green | North-North West | 1 | 2.30 m | 19.8 mm | 1.9 mm |
| 05/05/2010 | Green | Eastern residuals | 1 | 2.35 m | 84.1 mm | 4.4 mm |
| 07/11/2011 | Yellow | S then E residuals | 1 | 2.70 m | 55.2 mm | 4.2 mm |
| 22/05/2012 | Green | S then E residuals | 1 | 2.02 m | 44.1 mm | 2.7 mm |
| 20/01/2013 | Yellow | South-South West | 2 | 2.69 m | 50.3 mm | 3.1 mm |
| 12/02/2013 | Green | South-South West | 2 | 1.81 m | 38.3 mm | 2.3 mm |
| 30/03/2013 | Green | West | 1 | 2.06 m | 32.1 mm | 2.5 mm |
| 31/05/2013 | Yellow | North-North West | 1 | 3.25 m | 59.5 mm | 5.4 mm |
| 19/06/2013 | Yellow | South-South West | 1 | 2.71 m | 37.1 mm | 4.5 mm |
| 06/11/2013 | Green | North-North West | 1 | 2.01 m | 44.2 mm | 3.0 mm |
| 19/11/2013 | Yellow | Eastern residuals | 2 | 2.88 m | 83.4 mm | 5.2 mm |
| 25/01/2014 | Orange | North-North West | 1 | 3.78 m | 79.1 mm | 2.6 mm |
| 06/03/2014 | Green | North-North West | 1 | 2.15 m | 33.0 mm | 2.9 mm |
| 04/04/2014 | Green | South-South West | 1 | 2.35 m | 37.9 mm | 2.9 mm |
| 27/02/2015 | Yellow | North-North West | 3 | 2.92 m | 62.1 mm | 2.6 mm |
| 04/03/2015 | Green | North-North West | 2 | 2.36 m | 27.1 mm | 1.8 mm |
| 28/04/2015 | Green | S then E residuals | 1 | 2.00 m | 33.5 mm | 3.1 mm |
| 26/11/2015 | Green | North-North West | 1 | 2.34 m | 30.9 mm | 1.4 mm |
| 13/02/2016 | Green | North-North West | 1 | 2.28 m | 55.3 mm | 3.2 mm |
| 22/01/2018 | Green | North-North West | 1 | 2.12 m | 15.3 mm | 1.1 mm |
| 21/02/2018 | Green | North-North West | 1 | 2.02 m | 41.5 mm | 1.9 mm |
| 08/05/2018 | Green | South-South West | 1 | 2.44 m | 34.3 mm | 6.2 mm |
| 13/06/2018 | Green | North-North West | 1 | 2.28 m | 43.1 mm | 3.3 mm |
| 16/07/2018 | Yellow | South-South West | 1 | 2.56 m | 92.9 mm | 14.9 mm |
| 26/05/2019 | Green | North-North West | 1 | 2.20 m | 50.9 mm | 3.1 mm |
| 14/12/2019 | Yellow | North-North West | 1 | 3.49 m | 51.0 mm | 3.7 mm |
| 23/01/2020 | Green | Eastern residuals | 1 | 2.02 m | 26.1 mm | 4.6 mm |
| 22/04/2020 | Green | South-South West | 1 | 2.20 m | 53.8 mm | 2.6 mm |
| 16/05/2020 | Green | West | 2 | 1.92 m | 22.9 mm | 2.6 mm |
| 12/12/2020 | Green | North-North West | 1 | 2.21 m | 62.0 mm | 4.3 mm |
| 01/02/2021 | Green | North-North West | 1 | 2.31 m | 55.8 mm | 3.5 mm |
| 11/12/2021 | Yellow | North-North West | 1 | 2.95 m | 60.7 mm | 3.1 mm |
| 11/01/2022 | Orange | North-North West | 1 | 4.31 m | 77.0 mm | 4.3 mm |

## Appendix C: F6 and F8 full results

Figures C1 and C2 work as follows: on left-hand panels (a), (c), (e) are plotted scores per set, right-hand panels (b), (d), (f) show scores per event. The score is marked by a dot, its colour refers to the set it comes from: orange or brown for LDS, green or dark green for TDS. On the left panels, the line's color depends on the lowest score: a green line highlights a test score






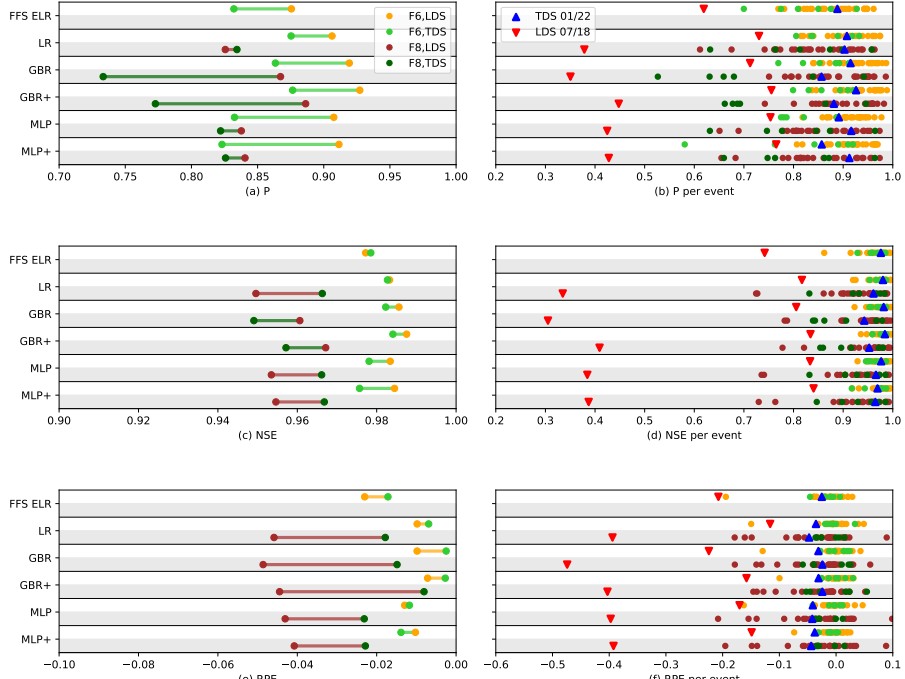

**Figure C1.** Performance scores on all models for sets and events in the F6,C and F8,C settings. The criteria are computed by sets in (a) and by events in (b). Orange (resp. red) dots are for LDS events in F6,C (resp. F8,C) and light (resp. dark) green dots for TDS events. On the right-hand panel, the 20-year event (Jan. 2022, TDS) is in blue upward triangle, in downward red triangle is the event with the biggest instantaneous rains (Jul. 2018, LDS).

under training score (common in ML), whereas a orange line points to a training score under the testing score (uncommon, hints at unexpected results). The upwards blue triangle on right-hand panels shows scores for the January 2022 event, with highest peak discharge (always in TDS). The downward triangle on right-hand panels shows scores for the July 2018 event, with highest peak rainfall (blue in deterministic TDS, red in chronological LDS).

Figures C3 and C4 work as follows: on left-hand panels (a), (c), (e) are plotted scores per set, right-hand panels (b), (d), (f) show scores per event. The score is marked by a symbol, its colour refers to the set it comes from: orange for LDS, green for TDS. On the left panels, the line's color depends on the lowest score: a green line highlights a test score under training score (common in ML), whereas a orange line points to a training score under the testing score (uncommon, hints at unexpected results). The upwards blue triangle on right-hand panels shows scores for the January 2022 event, in TDS, with highest peak discharge. The downward blue triangle on right-hand panels shows scores for the July 2018 event, in TDS, with highest peak rainfall.





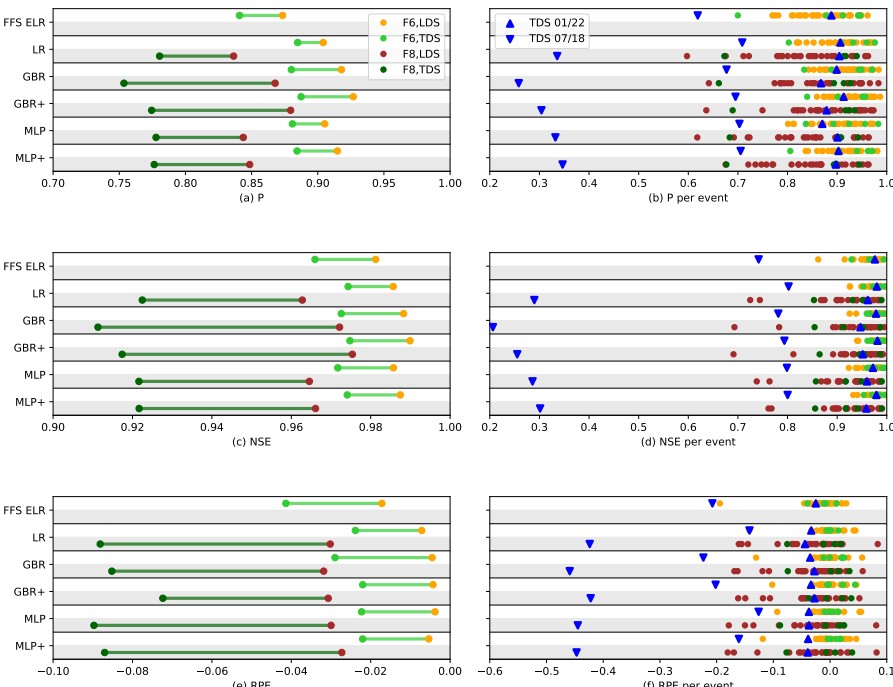

**Figure C2.** Performance scores on all models for sets and events in the F6,D and F8,D settings. The criteria are computed by sets on the left-hand panels and by events on the right-hand panels. Orange dots are for LDS events and green dots for TDS events of the F6 input settings. In the same fashion, brown stands for F8,LDS and dark green for F8,TDS. The 20-year event (Jan. 2022, TDS) is shown in blue upward triangle on the right-hand panels, in downward blue triangle is the event with the biggest instantaneous rains (Jul. 2018, TDS).





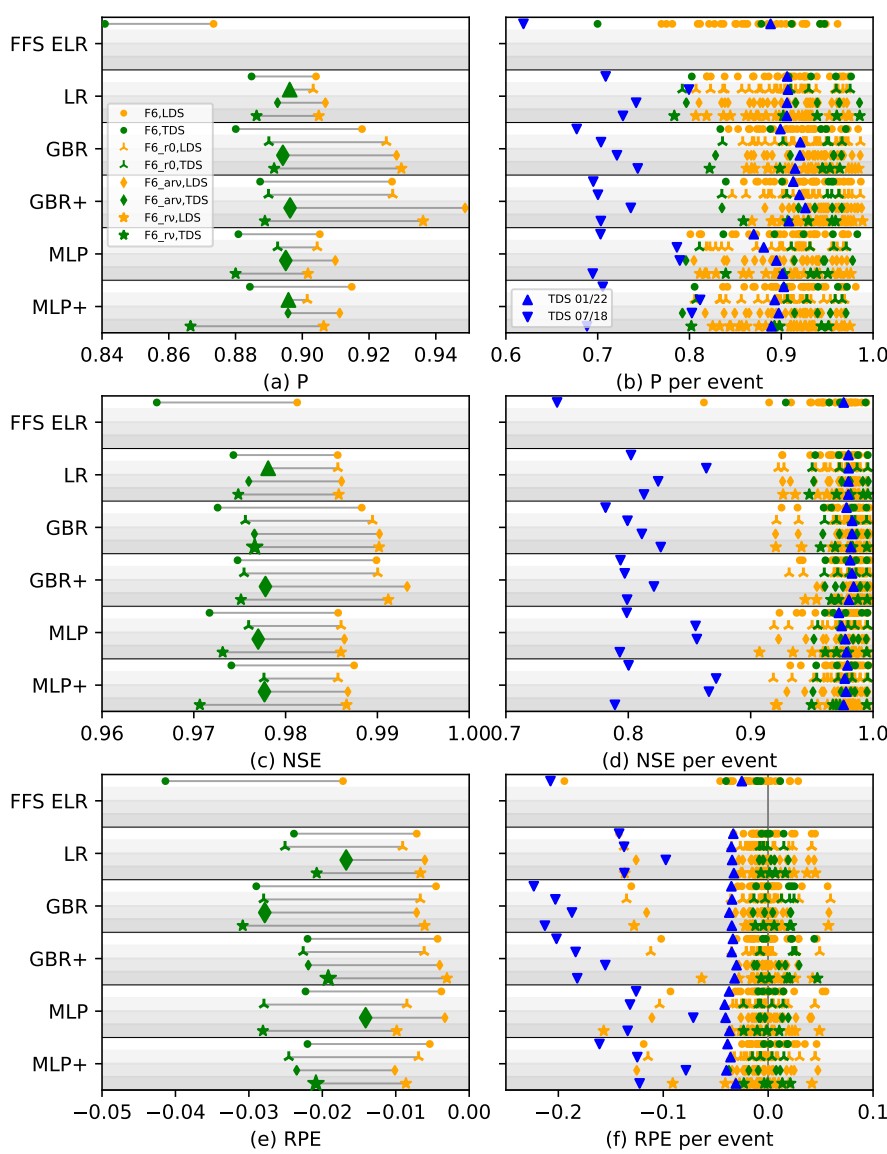

**Figure C3.** Scores for all models for configurations $F6$, $F6\_r0$, $F6\_ar\nu$ and $F6\_r\nu$ (resp. dot, triangle, diamond, star). Left (resp. right) panel: computed per set (resp. event). Orange (resp. green) symbols indicate scores for LDS (resp. TDS). The Jan. 2022 event is indicated as a blue upward triangle and the Jul. 2018 is indicated as a downward blue triangle. Both events are in TDS).




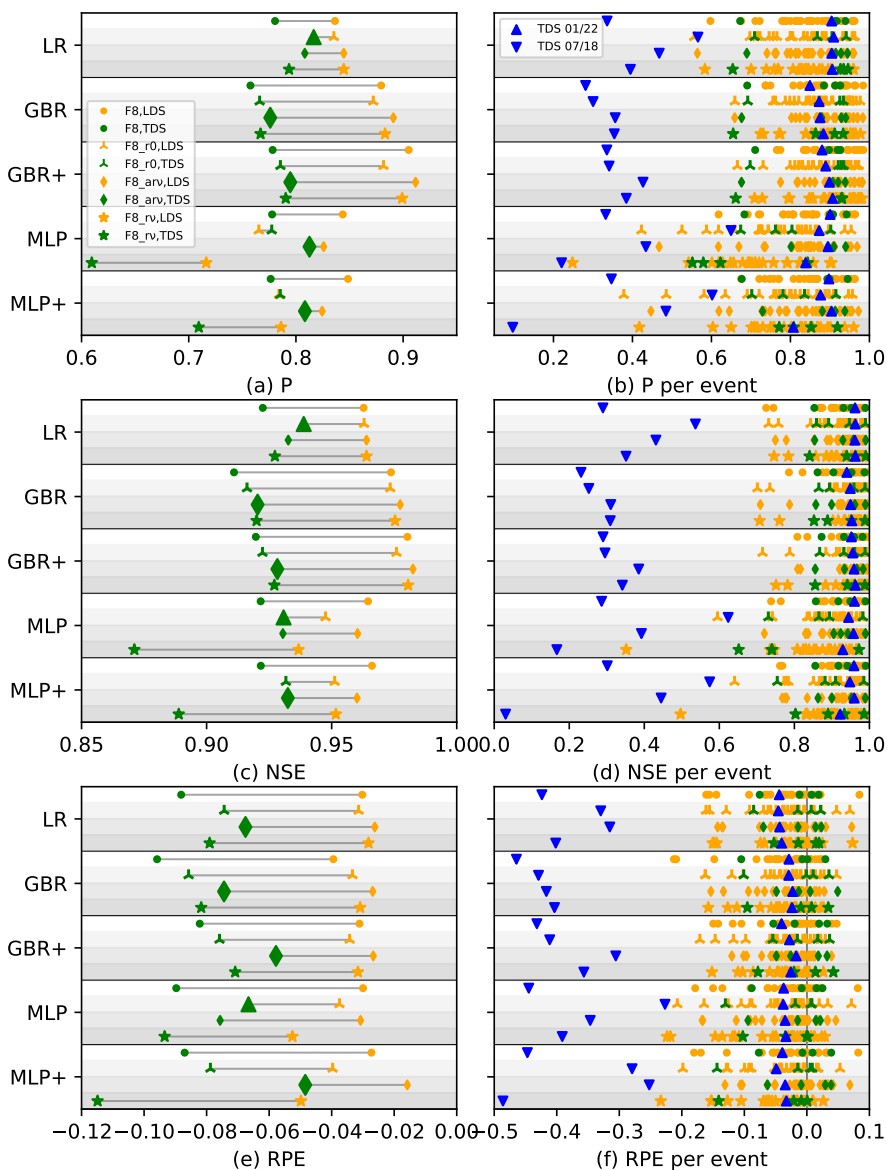

**Figure C4.** Scores for all models for configurations $F8$, $F8\_r0$, $F8\_ar\nu$ and $F8\_r\nu$ (resp. dot, triangle, diamond, star). Left (resp. right) panel: computed per set (resp. event). Orange (resp. green) symbols indicate scores for LDS (resp. TDS). The Jan. 2022 event is indicated as a blue upward triangle and the Jul. 2018 is indicated as a downward blue triangle. Both events are in TDS).



## Appendix D: Rain fall data time frame length selection

The models are trained with every time-frame $\nu \in [\![2, 24]\!]$ set as time-series, and also averaged over (as two different settings), to take longer timescales into account. To prevent over fitted models, the best results of each algorithm shown in sect. 4.2.1 and 4.2.2 were only selected with cross-validated (see sect. 3.2) LDS scores. As shown in sect. 3.3.3, the RPE is more sensitive to variations, thus it is best to prioritize and maximize both NSE and $P$ scores (best: 1) while only maintaining a reasonably low RPE (best: 0) score.

Selected values of $\nu$ for $F6\_ar\nu$ are $\nu_{\text{LR,F6\_ar}} = 17$, $\nu_{\text{GBR,F6\_ar}} = 5$, $\nu_{\text{GBR+,F6\_ar}} = 9$, $\nu_{\text{MLP,F6\_ar}} = 12$ and $\nu_{\text{MLP+,F6\_ar}} = 7$.

For $F6\_r\nu$, they are $\nu_{\text{LR,F6\_r}} = 13$, $\nu_{\text{GBR,F6\_r}} = 10$, $\nu_{\text{GBR+,F6\_r}} = 23$, $\nu_{\text{MLP,F6\_r}} = 2$ and $\nu_{\text{MLP+,F6\_r}} = 3$.

The selected values for $F8\_ar\nu$ are $\nu_{\text{LR,F8\_ar}} = 13$, $\nu_{\text{GBR,F8\_ar}} = 16$, $\nu_{\text{GBR+,F8\_ar}} = 11$, $\nu_{\text{MLP,F8\_ar}} = 13$ and $\nu_{\text{MLP+,F8\_ar}} = 13$.

For $F8\_r\nu$, they are $\nu_{\text{LR,F8\_r}} = 14$, $\nu_{\text{GBR,F8\_r}} = 8$, $\nu_{\text{GBR+,F8\_r}} = 6$, $\nu_{\text{MLP,F8\_r}} = 4$ and $\nu_{\text{MLP+,F8\_r}} = 2$.

*Author contributions.* T. Defontaine developed the model code and performed the simulations. Main writer: T. Defontaine, co-writers and proofreaders: S. Ricci, C.J. Lapeyre

*Competing interests.* The authors declare that no competing interests are present

*Acknowledgements.* The authors wish to thank the local Flood Forecasting Services (SPC Garonne Tarn Lot) and the SCHAPI for their help during this project, and without whom this work would not have been thoroughly conducted.



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
