# Peer review of "Real-time flood forecasting with Machine Learning using scarce rainfall-runoff data"

_EGUsphere, 2023_

## Author Comment (AC1)

Dear reviewers, editors,

The authors thank you for considering and reviewing our work entitled "Real-time Flood Forecasting with Machine Learning Using Scarce Rainfall-Runoff Data."

We greatly appreciate the extensive and positive comments provided by both reviewers. Their constructive feedback and insightful questions have been invaluable, and we have addressed each comment accordingly. These responses outline the significant changes we will make to the article based on the reviewers' suggestions.

Below, you will find a point-by-point response to the comments from Reviewer 2, with our answers highlighted in blue.

**Reviewer 2**

In terms of the selection of evaluation indicators, NSE (a common indicator for measuring the simulation accuracy of hydrological models), persistence index, which helps evaluate the stability of the model, and relative peak error (RPE) were selected as one of the indicators for evaluating the accuracy of flood peak simulation by hydrological models. These three indicators focus on the integrity, stability, and simulation accuracy of the flood peak, but do not consider the peak time of the flood peak. The flood peak occurrence error is an indicator that measures the difference between the simulated flood peak occurrence time and the actual observation data, which helps to evaluate the simulation accuracy of the flood peak occurrence time. Adding this indicator can more comprehensively evaluate the performance of the hydrological model in flood simulation.

This criterion could indeed be considered. However, our events mostly contain only one peak, and this criterion could also be too sensitive to noises and not yield any decisive information. Moreover, we believe that the conclusions will remain consistent as long as all events and models are compared using the same criteria (the ones that are currently in the paper).

---

## Author Comment (AC2)

Dear reviewers, editors,
The authors thank you for considering and reviewing our work entitled "Real-time Flood Forecasting with Machine Learning Using Scarce Rainfall-Runoff Data."

We greatly appreciate the extensive and positive comments provided by both reviewers. Their constructive feedback and insightful questions have been invaluable, and we have addressed each comment accordingly. These responses outline the significant changes we will make to the article based on the reviewers' suggestions.

Below, you will find a point-by-point response to the comments from Reviewer 1, with our answers highlighted in blue. We extend our sincere thanks to Reviewer 1 for their thorough review.

**reviewer 1**

General comments

The aim of this article is to implement and discuss the application of 3 data-driven models for flood forecasting on the Garonne River in Toulouse. The 3 models are a linear model and two types of Neural Networks: MLP and GBR. The title of the article mentions the fact that the available data are scarce, but this seems inaccurate to me. The database is derived from meteorological radar data and water level and flow data from several upstream stations, with a 1h time step, which seems to be a very satisfactory basis for designing such models.

The authors acknowledge that the Garonne catchment is well monitored, with extensive hourly hydrological and meteorological data available. However, data scarcity arises from focusing on extreme events, which, by definition, occur only a few times per year at most. Additionally, these extreme events vary in nature, being caused by diverse atmospheric and hydrologic conditions. In this study, the extreme events are categorized based on wind, the number of peaks, and peak intensity. Among the 36 events analyzed, some types are well represented, while others are represented by only a single event. Consequently, including these outlier events in the training or validation datasets significantly impacts the results and complicates prediction. Clarification will be provided in the abstract, introduction and Section 2.

The approach relies on a methodology based on choices that are not always well explained. Why radar and not raingauge data, how forecast horizons are chosen, why increment to flow and not flow is calculated, why neglect the rain upstream Toulouse, etc.

"why radar and not raingauge data"

As mentioned in Section 2.1, the ANTILOPE rainfall product combines radar observations with rain gauge measurements, providing the best rain analysis available for the French territory. This product is post-processed as a cumulative and weighted dataset for operational forecasting to provide the LAMEDO product, which is used in this study. Therefore, the study is not limited to using radar data alone. This will be rephrased for clarity.

"how forecast horizons are chosen"

The lead times for this predictive study were selected based on those used by the operational flood forecast service for the Garonne-Tarn-Lot area. Currently, this service issues forecasts for +4h and +6h lead times, which we have adopted in our study. Additionally, one objective of our research is to extend the methodology to longer lead times, such as +8h.

 "why increment to flow and not flow is calculated,"

The flood forecast service in Toulouse has developed and calibrated an empirical lag and route model for +4 and +6 hour lead times. This model operates by analyzing discharge anomalies between upstream observing stations and the outlet at Toulouse Pont Neuf (TPN). The discharge anomaly at TPN, calculated over the period from the present time to the targeted prediction lead time, is derived as a linear combination of discharge anomalies from upstream stations. To compare the performance of machine learning algorithms with the operational service, we have used similar inputs. Additionally, it is important to note that the relationship between upstream and downstream discharge anomalies is more linear than the relationship between the discharges themselves.

"why neglect the rain upstream Toulouse"

We believe there may be some confusion here by Reviewer 1. The purpose of incorporating the rainfall product is to account for the volume of water entering the catchment, which is not captured by discharge measurements at Vigicrue observing stations. Specifically, we consider rainfall between TPN and the stations just upstream of Toulouse.

> But the methodology used seems rigorous insofar as the validation and test sets are quite different, and hyperparameters are selected by cross-validation. The authors also make no secret of the intense event of 2018 on which the forecasts fail. The description of the database lacks cross-correlations between rainfall and discharge, or better still, discharge increases. This would help identify propagation or response times, and the significance of the rainfall used.

The cross-correlations between rainfall and discharges were made but did not yield any more information than what was already provided in the paper.

The quality criteria chosen: Nash and persistence are both used, which is good and rigorous. The EPR focuses on the peak, but it seems to me that measuring only the error on the peak's maximum value would be more selective than the proposed calculation, to avoid diluting the value of the peak's maximum in those of the peak as a whole. The results would, it seems to me, be very different.

This criterion could indeed be considered. However, we believe that the conclusions will remain consistent as long as all events and models are compared using the same criteria.

The main problem with the study is that all the models, whether linear or non-linear, produce virtually the same results, and that the latter are of no operational use, since the actual forecast horizon is closer to the hour than to the envisaged forecast horizon (4, 6, 8h).

The predictions are issued for +6h and +8h lead times, not +1h as Reviewer 1 seems to understand. The event of July 2018 is the only one that does not satisfy the envisaged forecast horizon. The data frequency is hourly, ensuring that the forecasting capability of the algorithms aligns with operational needs.

Furthermore, in the description of the models, it is necessary for each model to have the exact inputs applied. For example, it is clearly stated that upstream flows are applied, but is the flow at Toulouse also entered at time t?

The discharge anomaly at Toulouse Pont Neuf is the output of the algorithms, while the inputs consist of discharge anomalies from selected upstream stations. For some machine learning algorithms implemented in the study, the discharge anomaly forecasted by the flood forecasting service is used as an optional additional input; however, the primary focus remains on the predictions generated by the algorithms at Toulouse Pont Neuf from upstream discharges.

If so, we should try to remove it. Indeed, this information is so important to the model that it runs the risk of ignoring the rainfall, especially if the latter is not very well observed.

—

My feeling is that the article presents a negative result: the neural networks fail to forecast the flow of the Garonne at Toulouse. In itself, a negative result can be published, but an attempt must then be made to explain it, and it is this last phase of the work that is lacking in the article. The authors do try to indicate that over-fitting is the cause of this lack of performance, but without argument. Moreover, this hypothesis contradicts the proposal to use an LSTM to perform prediction, as the LSTM is far more complex than, say, the MLP.

The article evaluates the performance of the machine learning algorithms across all events for +6h and +8h lead times. It demonstrates that the predictive capability of these algorithms surpasses that of the flood forecast service for the +6h lead time (when the operational model is available), and shows promising results for the +8h lead time (for which no operational predictions are available for comparison). The learning strategy has proven to be highly effective. Additionally, incorporating rainfall data as an extra input has shown significant value, and the ML algorithms effectively handle the variability in discharge and rainfall data.

However, the algorithms struggled with predicting the 2018 event, which is notably different from other events in the database, even with rainfall data included. The authors may have overly emphasized this negative result, potentially overshadowing the positive outcomes. We will revise Section 4, "Results," as well as the conclusion, to better highlight these positive results and provide clearer clarification.

> I would therefore urge the authors to attempt a more precise analysis of the quality of the precipitation applied, in order to find out whether it is the latter that is the cause of the models' inability to make a correct forecast for the 2018 event.

You are correct: the event in July 2028 is driven primarily by rapid rainfall responses and is less influenced by hydrological processes. As previously mentioned, this event differs significantly from others in the dataset, making it challenging to predict with the available data. We will emphasize this point more clearly in the Results section.

> Specific comments
>
> LL 81-82 "Most strategies are based on Neural Network (NN) models such as Multilayer Perceptron (MLP) (Riad et al., 2004; Mosavi et al., 2018; Noymanee and Theeramunkong, 2019), which is a simple version of feed-forward neural networks". The sentence is not right. MLP can be also recurrent.

This comment is valid, and we will make the appropriate modifications accordingly.

> L85 " highly correlated networks". What does "highly correlated networks" means ?

The authors will reformulate

> LL 107 "and the flow is quasi-linear,". The flow in itself cannot be linear or not linear. It is important to pay great attention to the strict meaning of what we write.

This comment is valid, and we will make the appropriate modifications accordingly. What we intended to convey is that the discharge anomaly at Toulouse can be approximated as a linear combination of the discharge anomalies from upstream stations.

LI 89-90 "Recent publications also showed the use of advanced NN models such as Recurrent Neural Networks (RNNs) and more specifically Long Short-Term Memory networks (LSTMs)" In fact, recurrent networks have been proposed since the 90s, as has the LSTM.

This comment is valid, and we will make the appropriate modifications accordingly.

L 148 "It is here assumed that the effect of rainfall between these stations and Toulouse is negligible": A correlatory analysis between flow increase and rainfall between the upstream stations and Toulouse would be a good way of proving its validity.

Eq. 2.2. The coherence of equation 2.2. doesn't immediately strike me: why do we have t-n and positive times in the first line, and t+n and negative times in the third? It seems to me that both lines must be written in the same way.

This comment is valid, and we will make the appropriate modifications accordingly.

Results analysis

First of all, the criterion tables in Figure 2 show that the results are quite good, except for the July 2018 event. Indeed, when we look at the limnigram in figure 3, the result is far less satisfactory. All the forecast peaks are about 4 or 5 hours late. This is obviously not satisfactory for a 6-hour forecasting horizon especially knowing that the event is in the training database. The performance of the different models is not significantly different.

As previously noted, we will enhance the description of the positive results by including a limnigram for an event where the prediction performs well, in addition to the one for the July 2018 event. There are several instances where the ML predictions are even more satisfactory than those of the operational model, as demonstrated in Figure 2. Including a limnigram for these events will provide a clearer illustration of the model's effectiveness.

The authors express the view that the poor results could be due to over-fitting. However, to be able to draw any real conclusions, we would need to know the complexity of the PM: how many layers, how many neurons, how many parameters? This is important if the radar image is to be applied as is.

It is important to note that the radar image is not used directly. Instead, the LAMEDO rainfall product utilized here is a time series of the recalibrated (with raingauges) radar image, specifically pondered over a sub-catchment area.

We can see that as the forecast horizon increases, the offset of the forecast curve increases accordingly.

The diagnosis that can be made is that the model waits for the observed flow value to increase at its input before passing this increase on to its output. The model therefore relies essentially on its flow inputs and neglects rainfall inputs. This is the main drawback of the feedforward model. This could indicate that the rainfall is not of good quality, or that the rain is falling very close to Toulouse and that the response time is therefore very short for the 2018 event. It would be good to discuss this point in the article. The visualization of the rainfall on the graph does not remove the doubt because, unless I read it inattentively, we don't know exactly what it represents (the average over all pixels)? If this is the case, it can't help us to answer the previous question.

The rainfall in the plot is the data the models see as input (LAMEDO time series). This rain data shown on the plot is a spatially pondered time series stemming from radar images (which are recalibrated with rain gauges).

Moreover, rainfall is only applied to the model via an instantaneous value (of the entire radar image?) or via the average (over 24h -2h). My recommendation is therefore twofold:

The post-processings (instantaneous or averages) are only applied on the time series, not on the radar images.

- firstly, to calculate the cross-correlation between radar rainfall and flow increase, and secondly, to do the same calculation with rain gauge information (several to introduce spatialization). If rain gauges have a better correlation with flow increase, then we try to use them.

As explained before, the product we actually use here is a time series of rainfall data. This data is obtained by a spatial ponderation of the radar product over the sub-catchment. The radar product is obtained from radar images that are passed through a kriging method with spatially distributed rain gauges.

- To study the question of initialization, of the model, this doesn't seem to be decisive, as all the models have the same behavior even though they have different architectures. But this is important information.

The authors agree with the comment here that the models all have similar behavior, with different architectures, meaning that the initialization of the models is not a decisive factor. We have made some trials before fixing the seed of the models (thus the random initialization). However, we ultimately did not see anything decisive linked to the initialization process. We acknowledge that there is too little information on this subject and will clarify this within the third section (the methods section).

Technical corrections

L 152: "As the non-linearity of the flow". Flow alone is not linear or non-linear, but the upstream_flow-downstream_flow relationship can be. Please tell us if this is the relationship we're talking about;

The authors will reformulate.

Eq. 2.2. The coherence of equation 2.2. doesn't immediately strike me: why do we have t-n and positive times in the first line, and t+n and negative times in the third? It seems to me that both lines could be written in the same way. This is not a good way of differentiating between calculation with mean value and calculation without mean value.

This comment is valid, and we will make the appropriate modifications accordingly.

Table 1: Is Q or deltaQ used for input and output? Indeed, what is indicated in the table seems to contradict what is written in equation 2.1 where it is the DelatQ that is estimated? You also need to be careful with difference calculations: they amplify noise, unlike additions, which average it out.

The authors acknowledge that table 1 needs to be modified and improved towards a more readable state. Only deltaQ is used as Input and Output.

A difference is needed when what we want to work with is variations of a variable (it is a differentiation of sorts), instead of the variable itself. We are working with transitory phenomena, mostly visible with a differentiated variable (here Q). We do not want to average the noises as most of the transitory information is contained within. We understand your point though : what we do here enhances the significance of other, unwanted noises.

The use of acronyms such as TDS adds nothing to the reading and forces the reader to go back and look up the definitions on the previous pages. It would be better to spell out all names.

The authors will remove this acronym and spell this out.

LL 224-225,I do not understand the sentence: "The ML models are trained to predict an ensemble of discharge values in TPN, issued hourly at the targeted forecast lead time, for all events in the DB." In fact, the test cannot be the entire database containing the learning events.

The authors understand the confusion, and will rewrite this more clearly. The models make instantaneous predictions, one step at a time. We can also give them multiple steps to get an ensemble of predictions. These ensembles can be as follow: an event, or a set (training or testing).

The authors also acknowledge that the use of the word "ensemble" may be confusing here, when it is in fact a set of values.

Figure 3: it is july 2018 and not 2028.

The authors will correct this mistake.